# Peripheral helper-T-cell-derived CXCL13 is a crucial pathogenic factor in idiopathic multicentric Castleman disease

Takuya Harada [1,8], Yoshikane Kikushige[1,2,8], Toshihiro Miyamoto[3], Kazuko Uno[4], Hiroaki Niiro[1], Atsushi Kawakami[5], Tomohiro Koga [5], Koichi Akashi[1,2] ✉ & Kazuyuki Yoshizaki [6,7] ✉

Castleman disease (CD) is a rare lymphoproliferative disorder. Among subtypes of CD, idiopathic multicentric CD-not otherwise specified (iMCD-NOS) has a poor prognosis and its pathogenesis is largely unknown. Here we present a xenotransplantation model of iMCD-NOS pathogenesis. Immunodeficient mice, transplanted with lymph node (LN) cells from iMCD-NOS patients, develop iMCD-like lethal inflammation, while mice transplanted with LN cells from non-iMCD patients without inflammation serve as negative control. Grafts depleted of human CD3[+] T cells fail to induce inflammation in vivo. Upon engraftment, peripheral helper T (Tph) cells expand and levels of human CXCL13 substantially increase in the sera of mice. A neutralizing antibody against human CXCL13 blocks development of inflammation and improves survival in the recipient mice. Our study thus indicates that Tph cells, producing CXCL13 play a critical role in the pathogenesis of iMCD-NOS, and establishes iMCD-NOS as an immunoregulatory disorder.

Castleman disease (CD), a rare polyclonal lymphoproliferative disorder, is characterized by lymphadenopathy, hypergammaglobulinemia, hepatosplenomegaly, and systemic inflammation[1]. Among several subtypes of CD, idiopathic multicentric Castleman disease (iMCD) is distinct from other subtypes of CD, including unicentric CD (UCD), which involves only one anatomical site, human herpes virus-8 (HHV-8)-associated MCD (HHV-8-MCD), and polyneuropathy, organomegaly, endocrinopathy, monoclonal plasma cell disorder, and skin changes (POEMS)-associated MCD (POEMS-MCD). Importantly, iMCD can be further classified into two distinct subtypes: iMCD with thrombocytopenia, anasarca, fever/elevated C-reactive protein (CRP), reticulin myelofibrosis, renal dysfunction, and organomegaly syndrome (iMCD-TAFRO) and iMCD-not otherwise specified (iMCD-NOS)[1,2].

The etiology of iMCD remains largely unknown. Surgical resection of solitarily enlarged lymph nodes (LN) in patients with iMCD-NOS has been shown to improve the symptoms of CD, suggesting that cells residing within LN have been considered to play a crucial role in the pathogenesis of CD[3]. Previous studies identified a proinflammatory cytokine produced by LN cells, interleukin-6 (IL-6), as a critical mediator of systemic inflammation of iMCD[1,3]. Anti-IL-6 or anti-IL-6 receptor antibody therapies with siltuximab or tocilizumab have become the standard treatment for iMCD[4–6]. However, approximately half of iMCD patients do not respond to IL-6 inhibition. Therefore, alternative biological pathways are suspected to be involved in the pathogenesis of iMCD cases refractory to IL-6 inhibition[7,8]. A recent study identified CXCL13 as the most elevated chemokine in the "flare" phase of iMCD, suggesting an important role of CXCL13 in the exacerbation of iMCD[7].

[1]Department of Medicine and Biosystemic Science, Kyushu University Graduate School of Medicine, Fukuoka, Japan. [2]Center for Cellular and Molecular Medicine, Kyushu University Hospital, Fukuoka, Japan. [3]Department of Hematology, Faculty of Medicine, Institute of Medical Pharmaceutical and Health Sciences, Kanazawa University, Ishikawa, Japan. [4]Luis Pasteur Center for Medical Research, Kyoto, Japan. [5]Department of Immunology and Rheumatology, Division of Advanced Preventive Medical Sciences, Nagasaki University Graduate School of Biomedical Sciences, Nagasaki, Japan. [6]The Institute of Scientific and Industrial Research, SANKEN, Osaka University, Osaka, Japan. [7]Medical corporation of Tokushukai, Osaka, Japan. [8]These authors contributed equally: Takuya Harada, Yoshikane Kikushige. ✉e-mail: akashi.koichi.357@m.kyushu-u.ac.jp; kyoshi@sanken.osaka-u.ac.jp

CXCL13 is produced by follicular dendritic cells (FDCs) during the formation of germinal centers in LN and is known to be involved in the migration of B cells that express CXCR5, a receptor for CXCL13. In addition to FDCs, recent studies have identified pathogenic CD4[+]PD-1[high]CXCR5[-] T cells, named peripheral helper T (Tph) cells, as CXCL13-producing cells in autoimmune diseases such as rheumatoid arthritis (RA) and systemic lupus erythematosus (SLE)[9–11]. Tph cells promote the migration of B cells within inflamed tissues and induce B cell differentiation into antibody-producing cells[12,13]. In addition to cytokines and chemokines, recent studies have emphasized the significance of specific signaling pathways in the etiology of iMCD. In iMCD-TAFRO, enhanced mammalian target of rapamycin (mTOR) signaling activity in LN tissue was described in IL-6 blockade-refractory patients[14]. Furthermore, type1 interferon (IFN) signaling is involved in the pathogenesis of iMCD-TAFRO through increased Janus kinase-dependent mTOR activation[15]. Based on these findings, the efficacy of rapamycin (sirolimus), a specific mTOR inhibitor, against iMCD-TAFRO, has been investigated in clinical trials[16].

In contrast, pathogenesis of iMCD-NOS is largely unknown. This is presumably due to the low incidence of iMCD-NOS and lack of appropriate animal experimental models. In the research field of hematopoietic and leukemic stem cells, the use of patient-derived xenograft (PDX) models using immunodeficient mice has dramatically improved the understanding of normal and malignant hematopoiesis[17,18]. Furthermore, PDX models have enabled us to evaluate the efficacy of therapeutic strategies against human hematological malignancies in vivo, as previously reported[19,20]. In addition to hematological malignancies, recent studies employed PDX models in the research of autoimmune diseases[21,22].

Here, we report that xenotransplantation of human LN cells from patients with iMCD-NOS lead to fatal inflammation in vivo, recapitulating iMCD-like cachexia with hypergammaglobulinemia and chronic inflammation. Furthermore, our PDX models revealed a prominent elevation of human CXCL13 and expansion of CXCL13-producing human Tph cells in immunodeficient mice. Moreover, blockade of human CXCL13 using a neutralizing antibody significantly improved lethal inflammation in mice transplanted with iMCD-NOS LN cells.

## Results

### Xenotransplantation of LN cells from patients with iMCD-NOS developed fatal systemic inflammation in the recipient mice

First, we transplanted $1.0–3.0 \times 10^6$ LN cells from seven independent patients with reactive lymphadenopathy ($n = 3$) and lymphoma ($n = 4$) (C1, C2, C3, C4, C5, C6, and C7) and three independent patients with iMCD-NOS (P1, P2, and P3) into the 2.2 Gy irradiated NOD.Cg-$Prkdc^{scid}Il2rg^{tm1Wjl}$/Sz (NSG) mice via tail vein injection (Fig. 1a). Supplementary Data 1 summarizes the clinical characteristics of the patients with iMCD-NOS and control subjects used in the study. Supplementary Fig.1 shows the histopathology images of iMCD-NOS LN of each patient. LN cells from patients with iMCD-NOS caused cachexia and fatal systemic inflammation in the recipient mice and all the mice died within two months, whereas none of the NSG mice transplanted with control LN cells (control NSG mice) died in all the cases examined (Fig. 1b and Supplementary Fig. 2a). The NSG mice transplanted with LN cells from patients with iMCD-NOS (iMCD-NOS NSG mice) exhibited body weight loss with depilation of the whole body and vasodilation in the ear, which is indicative of the cachectic status of mice, resembling symptoms of iMCD (Fig. 1c). All iMCD-NOS NSG mice were euthanized for frailty at 10 weeks. In contrast, all control NSG mice survived for 10 weeks after transplantation. Flow cytometry analysis of spleen and bone marrow (BM) of the recipient mice demonstrated the engraftment and expansion of human(h)CD45[+] hematopoietic cells in iMCD-NOS NSG mice evaluated ($n = 16$ for spleen and $n = 13$ for BM), whereas almost all the control NSG mice failed in the reconstitution of human hematopoiesis in vivo ($n = 41$ for spleen and $n = 42$ for BM)

(Fig. 1d and Supplementary Data 2). Of note, three out of 42 control mice showed reconstitution of hCD45[+] cells (>5%) in vivo, none of these mice developed systemic inflammation. Hematoxylin-eosin (HE) staining of the spleen and liver of iMCD-NOS NSG mice revealed a diffuse infiltration of human lymphocytes (Fig. 1e and Supplementary Fig. 2b). Immunohistochemical staining confirmed the expansion and diffuse infiltration of human CD3[+] T cells and CD20[+] B cells in the spleen and liver of the iMCD-NOS NSG mice (Fig. 1e and Supplementary Fig. 2b). These results collectively suggest that xenotransplantation of LN cells from patients with iMCD-NOS causes fatal systemic inflammation, recapitulating the iMCD-NOS-like cachectic status in vivo.

### Xenotransplantation of LN cells from patients with iMCD-NOS resulted in the expansion of memory B cells and plasmablasts harboring the secretion potential of human gamma-globulin

Flow cytometry analysis revealed the expansion of hCD45[+] hematopoietic cells, including hCD3[+] T cells and hCD19[+] B cells, in the spleen of iMCD-NOS NSG mice (Fig. 2a, left panels). We observed the engraftment and expansion of CD4[+] T cells as well as CD8[+] T cells in the BM, spleen, and liver of iMCD-NOS NSG mice (Supplementary Data 2). Consistent with the development of iMCD-NOS-like disease, hCD10[-]hCD19[+]hCD20[+]hCD27[+]hCD38[dim] memory B cells and hCD19[+]hCD20[-]hCD27[+]hCD38[high] plasmablasts[23,24] expanded in the spleens of recipient mice (Fig. 2a, right panels and Supplementary Data 2). In the spleen of recipient mice, human memory B cells and plasmablasts exhibited significantly higher frequencies than the other B cells in iMCD-NOS NSG mice, whereas human B cells did not efficiently expand in control NSG mice (Fig. 2b and Supplementary Fig. 3a). PCR analysis for the rearrangement of immunoglobulin heavy chain (IGH) genes revealed that these expanded human CD19[+] B cells were polyclonal in all cases examined (Supplementary Fig. 3b).

Given that memory B cells and plasmablasts expanded in iMCD-NOS NSG mice, we tested whether these cells could secrete human gamma-globulin into the sera of recipient mice. For this purpose, LN cells from another patient with iMCD-NOS (P4) were xenotransplanted into three NSG mice, and the iMCD-like disease was reconstituted in mice. Consistent with systemic inflammation, iMCD-NOS NSG mice exhibited significant weight loss after xenotransplantation (Fig. 2c). Serum levels of human IgG, IgA, and IgE, which has been shown to be associated with the response to IL-6 inhibition[25], in iMCD-NOS NSG4 mice were significantly higher than those in control NSG mice (Fig. 2d). Furthermore, splenomegaly was observed in iMCD-NOS NSG4 mice as compared to control NSG mice (Supplementary Fig. 3c, d).

### T-B interaction was required for the development of iMCD-like disease in vivo

To examine which lineage of cells was responsible for the development of iMCD-NOS-like systemic inflammation in mice, we xenotransplant the same number of unmanipulated LN cells and T cell-depleted LN cells from two independent patients with iMCD-NOS (P2 and P5) (Fig. 3a). Xenotransplantation of unmanipulated LN cells from patients with iMCD-NOS resulted in cachexia with efficient reconstitution of human hematopoietic cells, whereas transplantation of T cell-depleted LN cells from identical patients failed to reconstitute human hematopoiesis in the spleen (Fig. 3b). Furthermore, we observed significantly higher human gamma-globulin production in NSG mice transplanted with unmanipulated LN cells in both cases (Fig. 3c for iMCD-NOS NSG2 and Fig. 3d for iMCD-NOS NSG5), whereas transplantation of T cell-depleted LN cells from the patients (P2 and P5) failed to produce polyclonal human gamma-globulin, IgG, IgA, and IgE (Fig. 3c, d). In addition, T cell depletion canceled the body weight loss of iMCD-NOS NSG2 (Supplementary Fig. 4a). Histological analysis revealed that iMCD-NOS NSG2 mice transplanted with unmanipulated LN cells exhibited the infiltration of inflammatory cells in the kidney and lung,

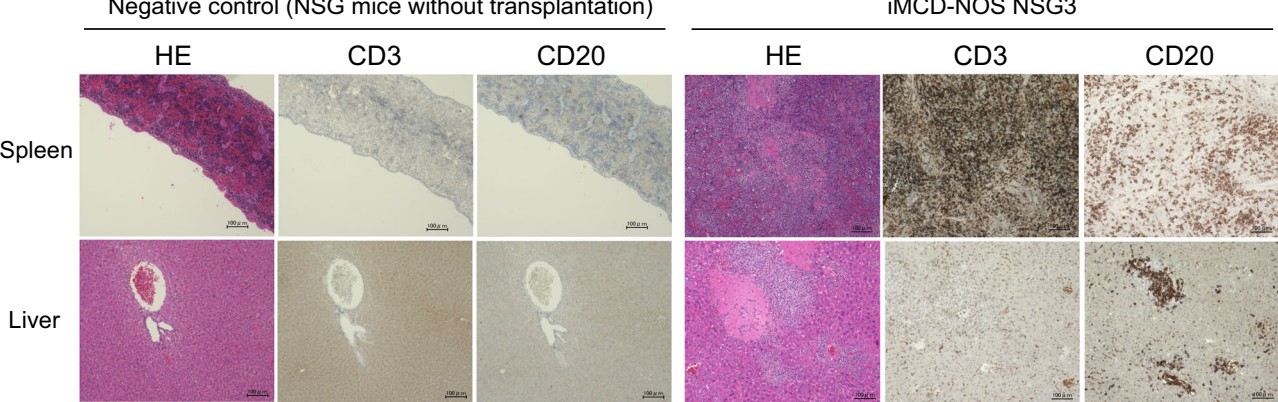

**Fig. 1 | Xenogeneic transplantation of iMCD LN cells developed fatal inflammation in vivo. a** A scheme for xenogeneic transplantation experiments is presented. **b** Survival analysis of recipient mice transplanted with LN cells from patients with iMCD-NOS (P1, P2, and P3: red) and control subjects (C1, C2, C3, C4, C5, C6, and C7: black). LN cells from identical patients were transplanted into at least six recipient mice. Log-rank test $P = 5.26 \times 10^{-14}$. **c** Representative photos of the iMCD-NOS NSG mice exhibiting body weight loss with depilation of the whole body (right) and vasodilation in the ear (left). **d** Frequencies of human CD45[+] hematopoietic cells in the spleen (left) and BM (right) of the recipient mice transplanted with control LN (black, spleen: $n = 16$, BM: $n = 41$) and iMCD-NOS LN cells (red, spleen: $n = 13$, BM: $n = 42$). Comparison between the two groups was analyzed with two-tailed unpaired $t$ test ($P = 7.518 \times 10^{-12}$ for spleen and $P = 0.004338$ for BM). **e** Histological analysis of the spleen and liver of NSG mice without transplantation as negative control (left) and iMCD-NOS NSG3 mice (P3) (right). Images were obtained from three independent negative control mice and from three iMCD-NOS NSG1-3 mice (total nine iMCD-NOS NSG1-3 mice). **$P < 0.01$, ****$P < 0.0001$. IHC scale bar = 100 μm. Source data are provided as a Source Data.

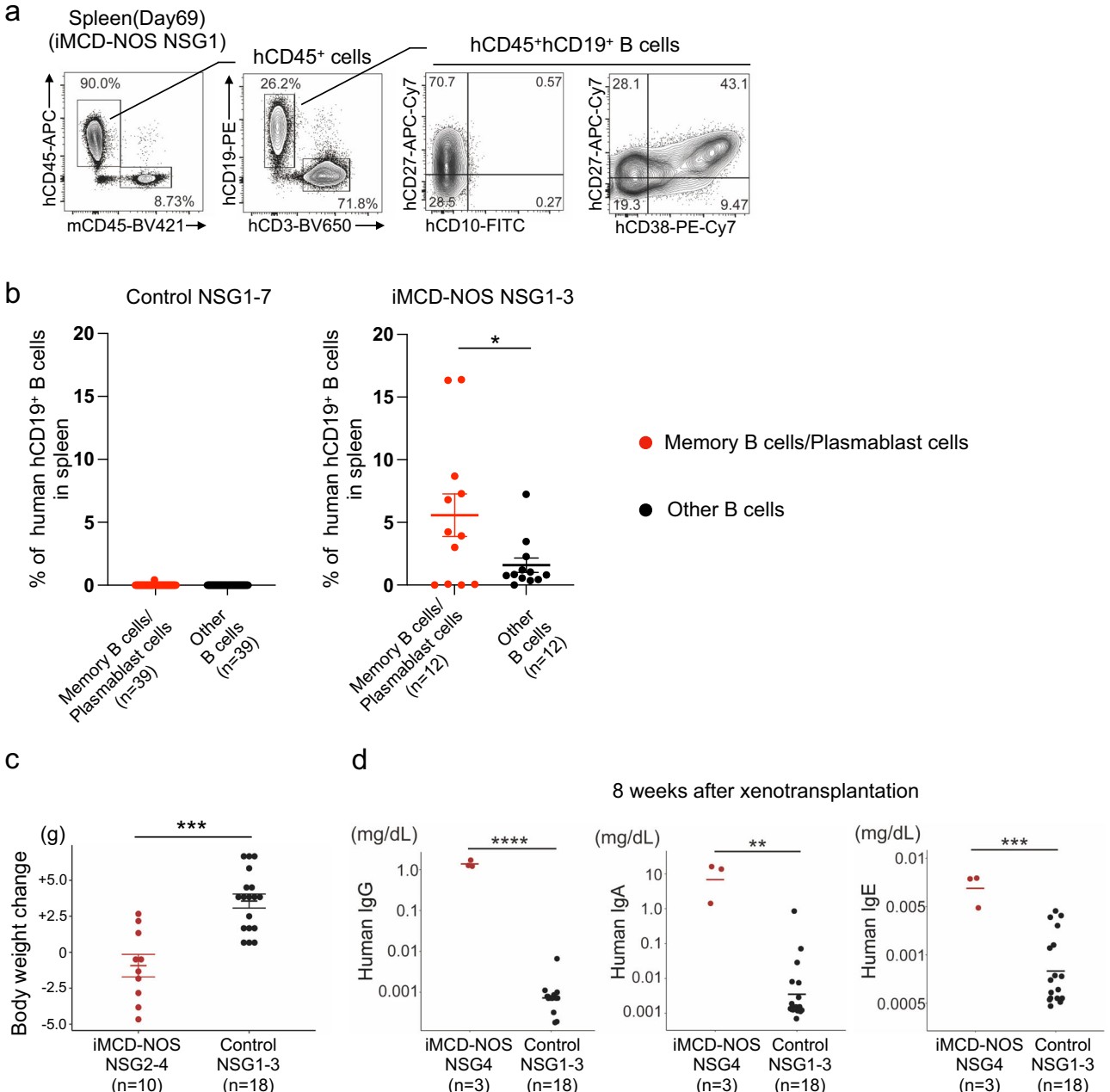

**Fig. 2 | The expansion of human plasmablast cells and the production of human gamma-globulin in iMCD-NSG mice. a** A representative FACS plot of cells in the spleen of iMCD-NOS NSG1 mice (P1). **b** Frequencies of human B cells (Memory B cells+Plasmablast cells, red vs. other B cells, black) in the spleen of the NSG mice transplanted with control LN (Control 1–7) cells ($n = 39$, $P = 0.3236$) and iMCD-NOS LN(P1-3) cells ($n = 12$, $P = 0.04406$). **c** Significant body weight loss in iMCD-NOS NSG2-4 mice (red, $n = 10$) after xenotransplantation of LN cells when compared to that in control NSG mice (black, $n = 18$). ($P = 0.0001683$) **d** Quantification of human IgG, IgA, and IgE in the sera of iMCD-NOS NSG mice transplanted with LN cells from patient 4 (red, left, $n = 3$) and control LN cells (control 1–3) (black, right, $n = 18$). (IgG: $P = 1.199 \times 10^{-15}$, IgA: $P = 0.002296$ and IgE: $P = 0.0009838$) Comparisons between the two groups were analyzed with two-tailed unpaired $t$ test. Error bar represented as mean ± SE. *$P < 0.05$, **$P < 0.01$, ***$P < 0.001$, ****$P < 0.0001$. Source data are provided as a Source Data.

possible target organs of iMCD-NOS, whereas iMCD-NOS NSG2 mice transplanted with T cell-depleted LN cells did not (Supplementary Fig. 4b). These results strongly suggest that the interaction between human B cells and T cells plays a critical role in the development of iMCD-like systemic inflammation and expansion of memory B cells and plasmablasts in our iMCD-NOS NSG mouse model. Consistent with the critical role of T-B interaction in vivo, we confirmed the co-localization of hCD4+ T cells, hCD8+ T cells, and hCD20+ B cells in the spleen and liver of iMCD-NOS NSG2 mice using the Hyperion multicolor imaging system (Fig. 3e).

## hCD4+hPD-1highhCXCR5-hCCR2+ Tph cells expanded in iMCD-NOS NSG mice

We found that human T cells residing within the LN of patients with iMCD-NOS play a pivotal role in the development of iMCD-like disease through the interaction with B cells in the recipient mice. We investigated the subsets of human T cells in mice. We observed the expansion of hCD3+hCD4+hPD-1highhCXCR5- Tph cells in the spleen of iMCD-NOS NSG mice (Fig. 4a). Since non-iMCD-NOS LN cells failed in the efficient engraftment and reconstitution of human hematopoiesis in vivo (Fig. 1d and Supplementary Data 2), we evaluated the frequencies of

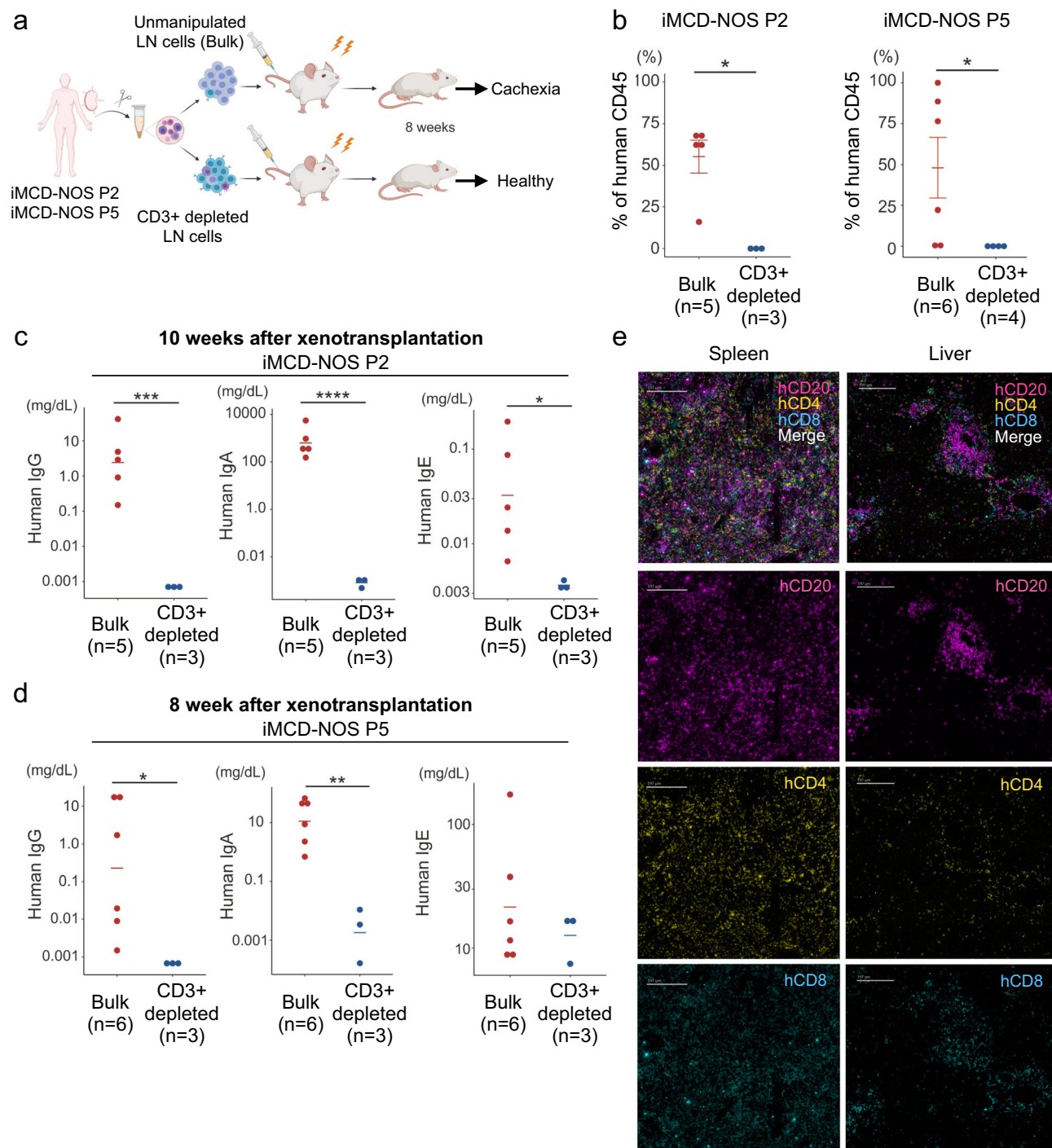

**Fig. 3 | T-B interaction was required for the development of iMCD-like systemic inflammation in vivo. a** A scheme for xenotransplantation experiments using unmanipulated bulk cells (upper panel) and CD3+ T cell-depleted LN cells (lower panel) from two independent patients with iMCD-NOS. **b** Evaluation of the frequency of human CD45+ hematopoietic cells in the spleen of recipient mice transplanted with unmanipulated (bulk) (left, red, P2: $n$ = 5 and P5: $n$ = 6) and CD3+ T cell-depleted LN cells (right, black, P2: $n$ = 3 and P5: $n$ = 4) at 8 weeks after xenotransplantation (P2: $P$ = 0.005023 and P5: $P$ = 0.04903). **c, d** Quantification of human IgG, IgA, and IgE in the sera of recipient mice transplanted with unmanipulated bulk LN cells (left, red, P2: $n$ = 5 and P5: $n$ = 6) and CD3+ T cell-depleted LN

cells (right, black, P2: $n$ = 3 and P5: $n$ = 3) from P2 (IgG: $P$ = 0.0009261, IgA: $P$ = 3.231 ×10−6 and IgE: $P$ = 0.02353) and P5 (IgG: $P$ = 0.005673, IgA: $P$ = 0.01775 and IgE: $P$ = 0.366). **e** Colocalization of human CD4+ T cells, CD8+ T cells, and CD20+ B cells in the spleen (left panels) and liver (right panels) of iMCD-NOS NSG2 mice, evaluated using the Hyperion Imaging System. We obtained the IMC images from at least three independent experiments. Error bar represented as mean ± SE. *$P$ < 0.05, **$P$ < 0.01, ***$P$ < 0.001. Comparisons between the two groups were analyzed with two-tailed unpaired $t$ test. IMC scale bar = 197 μm. Source data are provided as a Source Data.

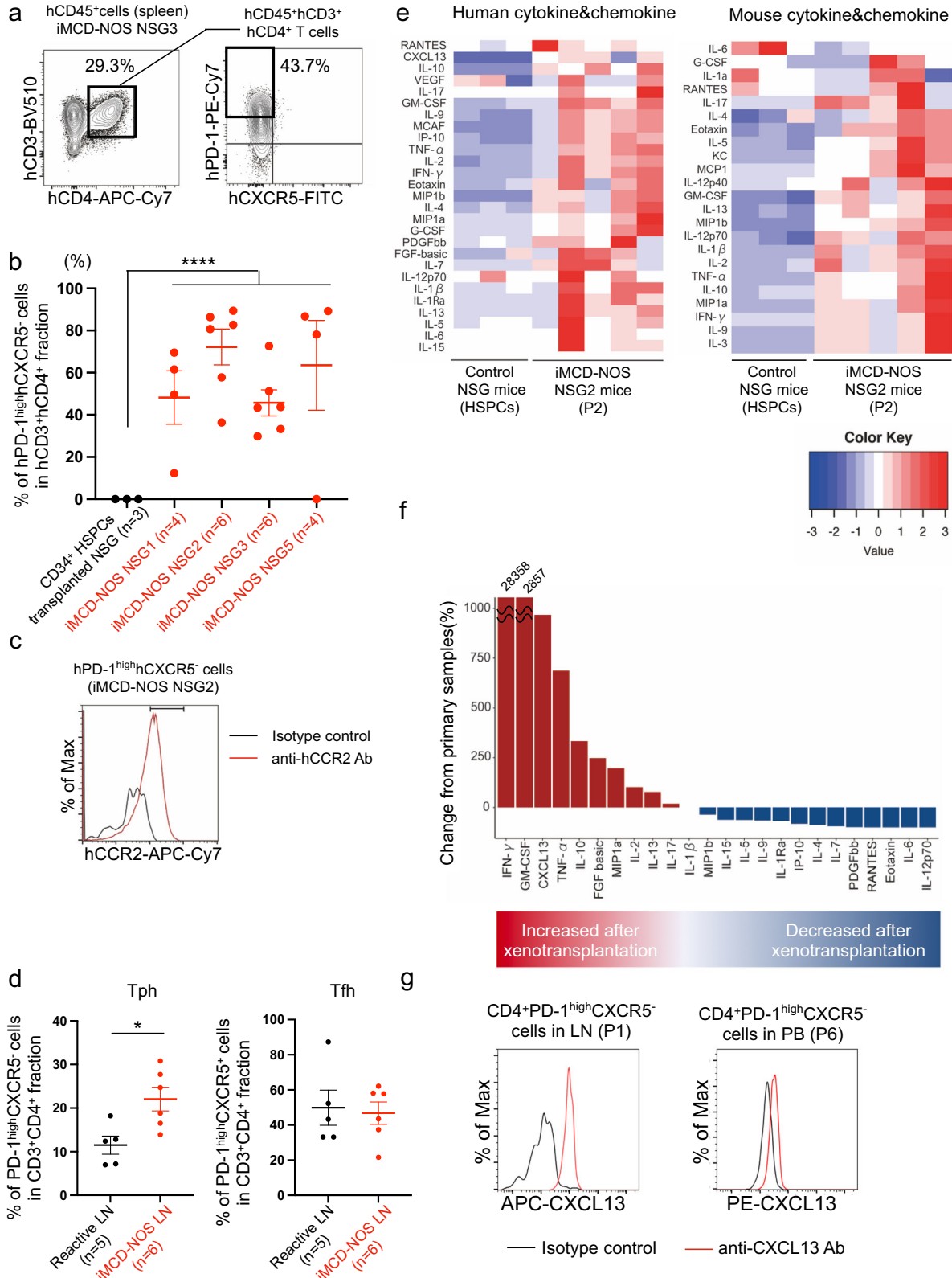

hCD3⁺hCD4⁺hPD-1^high^hCXCR5⁻ Tph cells in humanized NSG mice that reconstituted normal human hematopoiesis by xenotransplantation of cord blood (CB)-derived CD34⁺ hematopoietic stem/progenitor cells (HSPCs) as control subjects. The expansion of human Tph cells was observed in NSG mice transplanted with unmanipulated LN cells from all four independent patients with iMCD-NOS (P1, P2, P3, and P5) but not in humanized NSG mice (Fig. 4b and Supplementary Fig. 5b). The expanded hCD3⁺hCD4⁺hPD-1^high^hCXCR5⁻ cells expressed hCCR2, an

important marker for Tph cells[9] (Fig. 4c and Supplementary Fig. 5a). Furthermore, the frequencies of CD3⁺CD4⁺PD-1^high^CXCR5⁻ Tph cells in the iMCD-NOS LN cells (P4, P7, P8, P9, P10, and P11) were significantly increased as compared to those in the reactive LN cells (C1, C6, C7, C8, and C9) (Fig. 4d, left), whereas the frequencies of CD3⁺CD4⁺PD-1^high^CXCR5⁺ follicular helper T (Tfh) cells were not (Fig. 4d, right). It has been shown that Tph cells promote the maturation and expansion of B cells at the local site via the production of several cytokines and

**Fig. 4 | The expansion of hCD4⁺hPD-1^high hCXCR5⁻hCCR2⁺ Tph cells was specifically observed in iMCD-NOS NSG mice. a** A representative FACS plot of hCD45⁺ cells in the spleen of iMCD-NOS NSG3 mice. Human CD3⁺ CD4⁺ cells were analyzed for PD-1 and CXCR5 expression levels. **b** The frequencies of hPD-1^high hCXCR5⁻ cells among CD3⁺CD4⁺ human T cells in NSG mice reconstituted with normal human hematopoiesis via xenotransplantation of CB-derived CD34⁺ HSPCs (left, black, $n = 3$) and four independent iMCD-NOS NSG mice (right, red, NSG1: $n = 4$, NSG2: $n = 6$, NSG3: $n = 6$ and NSG5: $n = 4$). Two-group comparison between the iMCD group and the control group was performed using unpaired two-sided $t$ test ($P = 7.99 \times 10^{-9}$). Error bar represented as mean ± SD. **c** Representative FACS plot of hCCR2 expression in hPD-1^high hCXCR5⁻ Tph cells in the spleen of iMCD-NOS NSG2 mice (P2). **d** Comparison of the frequencies of CD3⁺CD4⁺PD-1^high CXCR5⁻ Tph cells and CD3⁺CD4⁺PD-1^high CXCR5⁺ Tfh cells in the reactive LN (C1, C6, C7, C8, and C9:

black, $n = 5$) and iMCD-NOS LN samples (P4, P7, P8, P9, P10, and P11: red, $n = 6$). Comparison between the two groups was analyzed with two-tailed unpaired $t$ test. (Tph: $P = 0.01358$ and Tfh: $P = 0.6635$) Error bar represented as mean ± SE. **e** Heatmap analysis of human (left) and mouse (right) cytokines and chemokines in the sera of NSG mice reconstituted with normal human hematopoiesis by xenotransplantation of CB-derived CD34⁺ HSPCs ($n = 3$) (left) and LN cells from Patient 2 ($n = 5$) (right). **f** Comparison of the levels of human cytokines and chemokines in the serum of the original patient (P2) and sera of iMCD-NOS NSG2 mice transplanted with LN cells from the same patient. **g** Representative intracellular staining of CXCL13 expression in CD4⁺PD-1^high CXCR5⁻ Tph cells from the LN of the patient with iMCD-NOS(P1) and PB of patient with iMCD-NOS (P6). **$P < 0.01$, ***$P < 0.001$, ****$P < 0.0001$. Source data are provided as a Source Data.

chemokines[9]. Therefore, we hypothesized that Tph cells play a critical role in the development of iMCD-like systemic inflammation through the production of several cytokines and chemokines.

## hCXCL13 as well as other cytokine/chemokines, including hIL-6, were produced by the engrafted human hematopoietic cells in iMCD-NOS NSG mice

The next aim was to determine which cytokines and chemokines are involved in the pathogenesis of iMCD. Therefore, we quantified human and mouse cytokines and chemokines in the sera of iMCD-NOS NSG2 mice transplanted with unmanipulated LN cells from the patient (P2) ($n = 5$) and NSG mice reconstituted with human normal hematopoiesis ($n = 3$) after transplantation of CB-derived CD34⁺ HSPCs, because the usage of humanized mice reconstituted with CB-derived CD34⁺ HSPCs as control subjects represented a standard method in PDX experiments[26–29]. iMCD-NOS NSG2 mice exhibited significant elevation of multiple human cytokines and chemokines in the sera, whereas NSG mice reconstituted with normal human hematopoiesis did not (Fig. 4e, left panel). Of note, the frequencies of hCD45⁺, hCD3⁺, and hCD19⁺ cells were not different between iMCD-NOS NSG2 mice and NSG mice reconstituting normal human hematopoiesis (Supplementary Fig. 5c). Murine cytokines and chemokines were also significantly elevated in iMCD-NOS NSG2 mice, reflecting the systemic fatal inflammation triggered by xenotransplanted LN cells from patients with iMCD-NOS (Fig. 4e, right panel). Among the elevated human cytokines from iMCD-NOS NSG2 mice, we sought to identify the cytokines and chemokines that could trigger systemic inflammation in iMCD-NOS NSG mice. For this purpose, we evaluated the alteration of cytokines and chemokines by comparing their serum concentrations in iMCD-NOS NSG2 mice and the original patient (P2). In iMCD-NOS NSG2 mice, IFN-γ, GM-CSF, CXCL13, and TNF-α levels were dramatically elevated compared to those in the serum of patient 2 (Fig. 4f). We also confirmed the elevated levels of IFN-γ, GM-CSF, CXCL13, and TNF-α in the serum obtained at the time of biopsy of LN (P2) as compared to those in the sera of healthy individuals ($n = 3$) (Supplementary Fig. 5d). All of these cytokines and chemokines are expressed in Tph cells[30,31]. Furthermore, a recent study revealed that CXCL13 is the most elevated chemokine during disease exacerbation in patients with iMCD-NOS[7], and we also confirmed the expression of CXCL13 in Tph cells in the LN of patient 1 and blood of patient 6 (Fig. 4g). In addition, we could not detect the expansion of other human CXCL13-producing cells including hCD45⁻hCD35⁺mouse(m)CD45⁻ FDC and hCD3⁺hCD4⁺hPD-1^high hCXCR5⁺ Tfh cells in iMCD-NOS NSG mice (Supplementary Fig. 5e). Based on these results, we hypothesized that CXCL13 produced by Tph cells plays a crucial role in the development of iMCD-like systemic inflammation at least in our xenograft model.

## Anti-hCXCL13 antibody treatment significantly improved the survival of iMCD-NOS NSG mice

To test whether CXCL13 is involved in the pathogenesis of iMCD-NOS, we conducted therapeutic experiments using our iMCD-NOS NSG mice. Unmanipulated LN cells from two independent patients with iMCD-NOS (P10 and P11) were transplanted into NSG mice. Figure 5a shows the experimental model. Mice were randomly assigned to the control and treatment groups 24 h after xenotransplantation. Control mice were intraperitoneally injected with mouse isotype IgG antibody (15 μg/mice) once a week for 4 weeks. The NSG mice in the treatment group were treated daily with an intraperitoneal injection of rapamycin (sirolimus) (40 μg/mice) for 4 weeks, or an intraperitoneal injection of anti-hCXCL13 mouse IgG1 antibody (15 μg/mice), which could neutralize the chemotactic activity of human CXCL13, once a week for 4 weeks.

We first performed therapeutic experiments using iMCD-NOS LN cells (P10) and found that anti-hCXCL13 antibody treatment significantly improved the survival of iMCD-NOS NSG10 mice (red line) compared to that of the control group (black line) (Fig. 5b). Figure 5c shows the survival analysis of iMCD-NOS NSG11 (P11) cells treated with isotype mouse IgG (black line), rapamycin (blue line), and anti-hCXCL13 mouse antibody (red line). Control mice developed fatal inflammation and most of them died within 20 days. Rapamycin treatment significantly improved the survival of iMCD-NOS NSG mice, where half of the mice survived within the observation period. Interestingly, anti-hCXCL13 antibody treatment completely blocked the development of fatal inflammation, and none of the iMCD-NOS NSG11 mice died. Although we could not perform statistical analysis due to the limited number of available serum samples at 8 weeks after transplantation, we observed the elevated serum albumin level in the NSG mice treated with anti-hCXCL13 antibody (Supplementary Fig. 6a). We finally evaluated the therapeutic effect of anti-hCXCL13 antibody using the LN samples obtained from the patient who exhibited the exacerbation during the treatment with IL-6R inhibitor (tocilizumab) (iMCD-NOS12) (Supplementary Data 1). The NSG mice in the treatment group were treated with an intraperitoneal injection of interleukin-6 receptor antibody (tocilizumab) (100 μg/mice), or anti-hCXCL13 mouse antibody (15 μg/mice) once a week for 6 weeks (Fig. 5d). There was no significant survival difference between control group and tocilizumab group, while there was a survival benefit with anti-hCXCL13 antibody treatment (Fig. 5e). We confirmed the expression of hCXCL13 in hCD3⁺hCD4⁺hPD-1^high hCXCR5⁻ human Tph cells in iMCD-NOS NSG12 mice (Supplementary Fig. 6b).

Collectively, these results suggest that hCXCL13, presumably secreted from Tph cells, plays a pivotal role in the promotion of systemic inflammation reproduced in our iMCD-NOS NSG mice by interacting with human B cells.

## Discussion

In the current study, we established a PDX model of iMCD-NOS using NSG mice, and revealed a critical role of the interaction between T and B cells in the pathogenesis of iMCD-NOS. We further identified a prominent elevation in human CXCL13 levels in the serum of recipient mice. Consistent with the significant elevation of CXCL13 levels, Tph cells harboring the potential for CXCL13 production expanded in vivo.

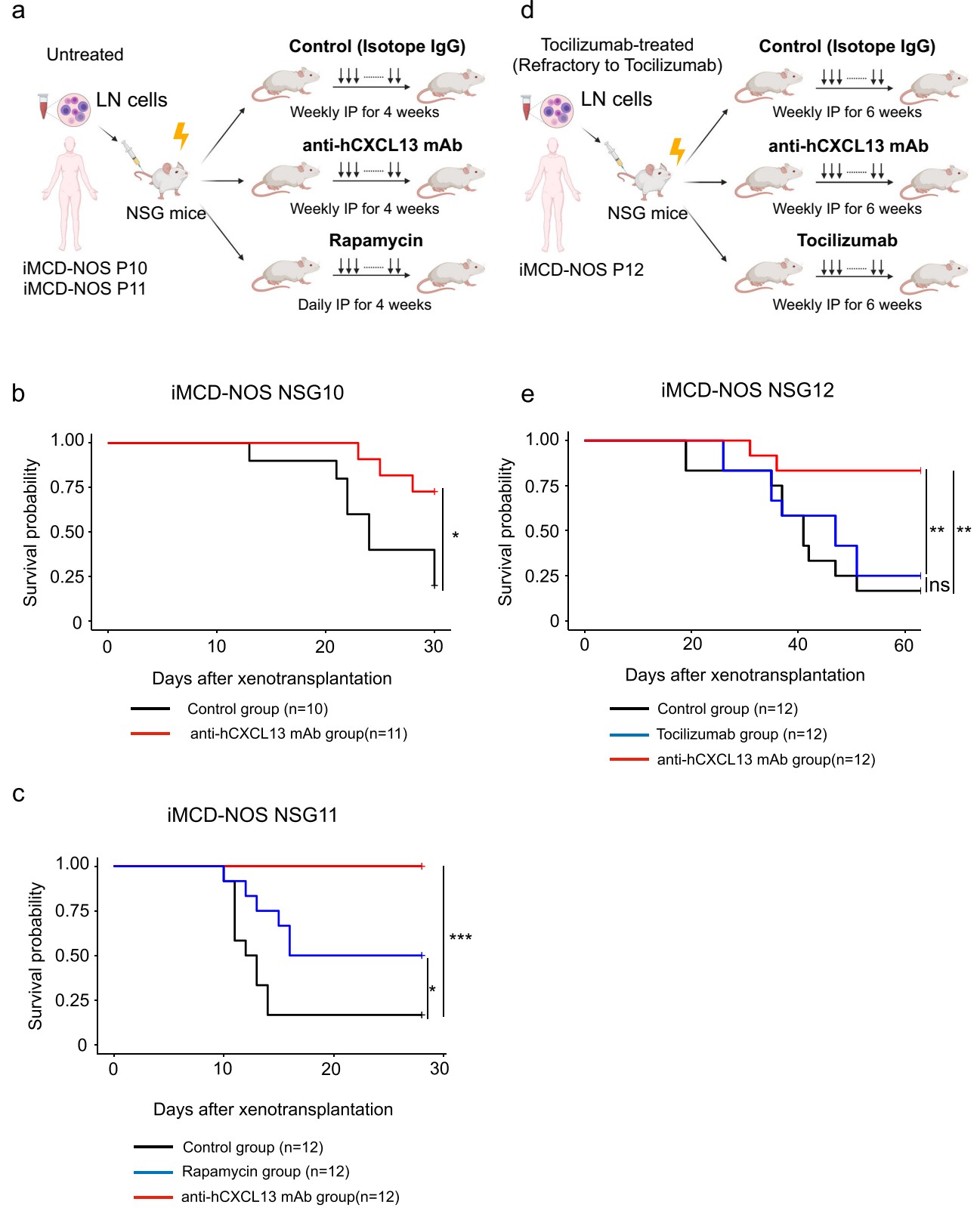

Finally, we have clearly shown that neutralization of CXCL13 using an anti-hCXCL13 monoclonal antibody dramatically suppressed lethal inflammation and improved the survival of iMCD-NOS NSG mice in vivo. Collectively, these results suggest that CXCL13 produced by Tph cells plays a significant role in the development of iMCD-like systemic inflammation in iMCD-NOS NSG mice, providing a rationale for CXCL13-targeting therapeutic applications in the treatment of iMCD-NOS. The proposed pathogenesis of iMCD-NOS provided in our study is summarized in Fig. 6.

As the depletion of human T cells from grafts completely abolished the development of iMCD-NOS-like systemic inflammation in vivo, our study indicated that the pathogenesis of iMCD-NOS depends on the abnormal activation of chemokine/cytokine-mediated immune cell crosstalk. Thus, iMCD-NOS represents an abnormal

**Fig. 5 | Anti-hCXCL13 antibody treatment significantly improved the survival of iMCD-NOS NSG mice. a** A scheme for the in vivo therapeutic experiments by xenotransplantation of LN cells from two independent patients with iMCD-NOS (P10 and P11) who were not treated before. Recipient mice were randomly assigned to control and treatment groups. After 24 h post-xenotransplantation, weekly intraperitoneal injection of mouse isotype IgG or anti-hCXCL13 antibody or daily intraperitoneal injection of rapamycin was initiated. **b** Survival analysis of iMCD-NSG10 mice treated with isotype mouse IgG ($n = 10$) (black) and anti-hCXCL13 antibodies ($n = 11$) (red). Log-rank test $P = 0.0133$. **c** Survival analysis of iMCD-NSG12 mice treated with isotype IgG ($n = 12$) (black), rapamycin ($n = 12$) (blue), and anti-hCXCL13 antibodies ($n = 12$) (red). Log-rank test $P = 0.0296$ (isotype IgG vs rapamycin) and $P = 0.0000314$ (isotype IgG vs anti-hCXCL13 antibodies). **d** A scheme for

the in vivo therapeutic experiments by xenotransplantation of LN cells from the patient with iMCD-NOS (P12) who exhibited the exacerbation during the treatment with IL-6R inhibitor (tocilizumab). Recipient mice were randomly assigned to control and treatment groups. After 24 h post-xenotransplantation, weekly intraperitoneal injection of mouse isotype IgG, tocilizumab, or anti-hCXCL13 antibody was initiated. **e** Survival analysis of iMCD-NSG12 mice treated with isotype mouse IgG ($n = 12$) (black), tocilizumab ($n = 12$) (blue), and anti-hCXCL13 antibodies ($n = 12$) (red). Log-rank test $P = 0.495$ (isotype IgG vs tocilizumab) and $P = 0.00259$ (isotype IgG vs anti-hCXCL13 antibodies). *$P < 0.05$, **$P < 0.01$, ***$P < 0.001$, ns not significant. Source data are provided as a Source Data.

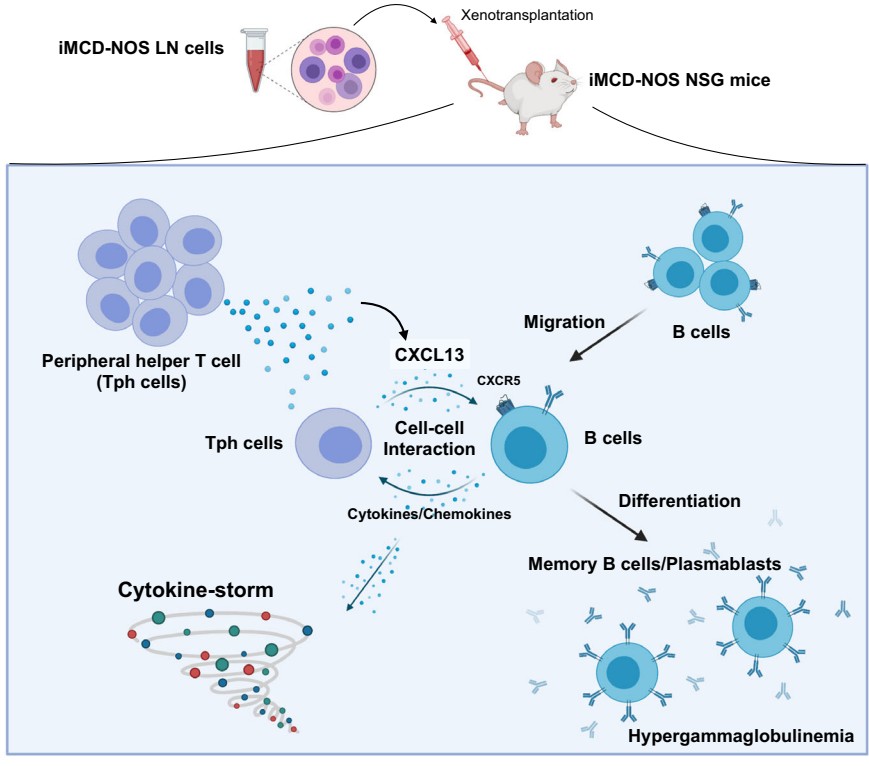

**Fig. 6 | A schematic summary of the proposed pathogenesis of iMCD-NOS in our PDX model.** Our xenotransplantation experiments revealed that the T-B interaction is required for the development of iMCD-like systemic inflammation in vivo with the elevation of human gamma-globulin. Aberrantly expanded CD4+PD-1highhCXCR5-CCR2+ Tph cells actively secrete CXCL13 and promote the migration of B cells toward inflamed sites. Through interaction with Tph cells, migrated B cells

differentiate into plasmablasts and initiate the secretion of human gamma-globulin, cytokines, and chemokines, leading to systemic inflammation. Thus, iMCD-NOS represents an abnormal immunoregulatory disorder in which the aberrant immunological synapse consisting of T-B interactions plays a critical role in disease initiation and exacerbation.

immunoregulatory disorder in which aberrant T-B interaction triggers systemic inflammation. Our data showed the essential role of T-B interaction and the co-localization of human T and B cells in mice for the development of iMCD-NOS like inflammation. Due to the limited cellular numbers of clinical samples available, we could not test the effect of B cell depletion in vivo, further studies including B cell depletion would be required to clarify the detailed mechanisms underlying T-B interaction in iMCD-NOS NSG mice.

Among these immune cellular components, we identified CD4+PD-1highhCXCR5- Tph cells as a critical subset involved in the pathogenesis of iMCD-NOS. Tph cells were originally identified as pathogenic CD4+ T cells that expanded in the joint tissues of patients with RA and SLE[9]. Recent studies have clarified the pathogenic function of Tph cells in the immunoregulation of diseases by promoting the migration of B cells within inflamed tissues and inducing B cell differentiation into antibody-producing cells[12,13]. CD4+PD-1highhCXCR5+ Tfh cells mediate B cell class switching and affinity maturation of

antibodies in the germinal center of lymphoid organs[12,32], whereas CD4+PD-1highhCXCR5- Tph cells are involved in the formation of ectopic lymphoid-like structures (ELSs) by secreting CXCL13 and promoting the migration of B cells in the inflammatory sites[9–11,32,33]. ELSs are involved in the pathogenesis of autoimmune diseases such as RA and SLE by promoting pathological immune responses[34–36]. Given that iMCD-NOS is characterized by multiple lymphadenopathies and sometimes involves ectopic extranodal sites such as the skin[37,38], lung[39–41], and kidney[41,42], aberrantly expanded Tph cells might be involved in the pathogenesis of iMCD-NOS through the formation of ELSs in the target organs. Since the expansion of human Tph cells was exclusively observed in iMCD-NOS NSG mice in our study, the aberrant differentiation toward Tph cells programmed in T cells of patients with iMCD-NOS might be one of the critical clues for understanding the pathogenesis of iMCD-NOS. Further studies are required to clarify the molecular mechanisms underlying the induction of aberrant differentiation and expansion of Tph cells during iMCD-NOS development.

Among cytokines and chemokines such as IFN-γ, GM-CSF, CXCL13, and TNF-α, which are secreted by Tph cells[30,31], our study demonstrates that CXCL13 represents a possible therapeutic target for the treatment of iMCD-NOS. Recent studies identified CXCL13 as the chemokine that is the most upregulated during the flare phase of iMCD[7] and important biomarker predicting efficacy of IL-6 inhibition in iMCD[43]. Consistently, the current study identified CXCL13 as a driving chemokine for iMCD-like inflammation in iMCD-NOS NSG mice. CXCL13 was originally identified as a B cell-specific chemoattractant[44,45]. CXCL13 plays a crucial role in lymphoid neogenesis and immune responses by binding to its receptor CXCR5. It has been shown that the CXCL13/CXCR5 signaling axis is involved in the pathogenesis of several autoimmune and inflammatory disorders[46]. Therefore, targeting the CXCR5 signaling pathway may be an alternative therapeutic strategy against iMCD-NOS.

The current study also offers an experimental iMCD-NOS NSG mouse model for studying the pathogenesis of iMCD. In general, PDX experiments have been mainly utilized in stem cell assays, including hematopoietic stem cells, leukemic stem cells, and cancer stem cells[47]. While the reconstitution of human mature lymphoid malignancies using primary samples remains challenging without specific modifications such as local injection into implanted human bones, flanks, omentum, or under the kidney capsule in PDX models[48–51], our mouse model stands out due to its lack of need for such modifications. Therefore, the current study provides a straightforward approach to reproduce non-malignant diseases in immunodeficient mice. The lack of appropriate experimental mouse models has hampered further investigation of the pathogenesis of iMCD-NOS, and our iMCD-NOS NSG mice represent a useful approach to recapitulate iMCD-NOS in vivo. Furthermore, PDX models enable us to evaluate the efficacy of therapeutic drugs, as shown in the current study. Therefore, studies using PDX experiments can contribute to the development of therapeutic approaches against iMCD-NOS.

However, there are several limitations in the PDX model. First, iMCD-NOS is a heterogeneous disease, and there is variability in the phenotype of reconstituted mice. We observed differences in the survival of mice and reconstitution of human hematopoietic cells in each transplanted case, implying the variability and cellular heterogeneity of transplanted iMCD-NOS LN cells. Furthermore, individual differences in mice can also affect the phenotype of the recipient mice. Second, the effect of xenogeneic graft versus host disease (GVHD) cannot be completely ruled out. Models of xenogeneic GVHD had been established via transplantation of higher number of $(1–5 \times 10^7)$ of human peripheral blood mononuclear cells[52]. In contrast, we transplanted less number $(0.6–3.0 \times 10^6)$ of iMCD-NOS LN cells. Thus, the amount and origin of human cells differ in the previous xenogeneic GVHD models. At least in our experiments, control LN $(1.0–3.0 \times 10^6)$ cells did not efficiently reconstitute human hematopoiesis in vivo, showing the distinct cellular characteristics of iMCD-NOS LN cells from the control subjects. Although the sample size was limited, three out of 42 mice transplanted with control LN cells and three NSG mice transplanted with CD34+HSPCs did not develop systemic inflammation and survived for a long term in the current study. Furthermore, PDX models of human mature lymphoid malignancies with specific modifications as described above has been shown to survive enough for the evaluation of human lymphoid malignancies without lethal inflammation in vivo[48–51]. Thus, the engraftment and expansion of human blood cells do not always lead to xenogeneic GVHD in vivo, and the induction of systemic inflammation seems to be specific to iMCD-NOS NSG mice. Furthermore, the impairment of B cell development and differentiation represents an important characteristic of GVHD[53–56]. Conversely, the engrafted human B cells expanded and preferentially differentiated toward memory and plasmablasts in iMCD-NOS NSG mice. We, therefore, considered that our PDX model recapitulated iMCD-like inflammation rather than GVHD at least in terms of the B cell expansion and differentiation. Third, human Tph cells expanded in

iMCD-NOS NSG mice, whereas FDC and Tfh cells did not. We, therefore, focused on Tph cells as the cellular source of CXCL13 at least in our PDX model, and these data did not exclude the possibility of FDC and Tfh cells as the sources of CXCL13 in the patients with iMCD-NOS. Further studies would be necessary to clarify the cellular sources of CXCL13 in iMCD-NOS. Even with these limitations, the development of PDX model represents a new approach to understanding the pathogenesis of iMCD-NOS.

In summary, we developed a PDX model that recapitulates human iMCD-NOS. Our PDX model demonstrated that CXCL13-producing Tph cells play a critical role in the pathogenesis of iMCD-NOS through their interaction with B cells, indicating that iMCD-NOS is an abnormal immunoregulatory disorder. Finally, this study has identified neutralization of CXCL13 to be a therapeutic approach against iMCD-NOS.

## Methods

### Human samples
Swelling lymph nodes that were surgically resected for diagnosis were obtained from seven control patients who were diagnosed with reactive ($n = 3$), classical Hodgkin lymphoma (cHL) ($n = 1$), plasmablastic lymphoma (PBL) ($n = 1$), diffuse large B-cell lymphoma (DLBCL) ($n = 1$), or follicular lymphoma (FL) ($n = 1$), and eleven patients who fulfilled the criteria for iMCD-NOS[1,2]. Supplementary Data 1 summarizes the clinical characteristics of the patients with iMCD-NOS and control subjects used in the study. Supplementary Data 1 also describes the CHAP score[57] for the evaluation of disease activity. Supplementary Fig. 1 shows the available histopathology images (HE staining) of surgically resected LN of each iMCD-NOS patient used in the study. The lymph nodes were minced into $1–3 \text{ mm}^3$ fragments in RPMI 1640 supplemented with 10% fetal bovine serum (FBS). The floating cells were collected from the medium. The cells were counted using Celltac Alpha MEK-6510K (Nihon Kohden) and subjected to subsequent analysis or cryopreserved as functional assay targets. Cord blood (CB) cells were obtained from full-term deliveries (Red Cross Blood Center, Japan Red Cross Society). Informed consent was obtained from all patients and controls in accordance with the Helsinki Declaration of 1975, revised in 1983. The institutional review boards of the participating hospitals, including Kyushu University Hospital (2020-576) and Tokushukai Hospitals (TGE01887-004), approved all research on human subjects and obtained patient permission from each hospital.

### Mice and xenotransplantation of human LN cells
Female NSG (NOD.Cg-*Prkdc^scid^Il2rg^tm1Wjl*/Sz, Jackson Labs, Strain#005557) mice at 4–6 weeks old were purchased from Charles River Laboratories Japan and Jackson Laboratory Japan. NSG mice were bred and housed in a specific pathogen-free facility in micro-isolator cages at Kyushu University. Animal experiments were performed in accordance with institutional guidelines approved by the Animal Care Committee of Kyushu University. The following conditions were controlled: daily light period from 07:00 to 19:00 and temperature at $23 \pm 1 \,°C$ with free access to water and food. For the reconstitution assays, selected cells (total $0.6–3.0 \times 10^6$ cells per mouse) in RPMI 1640 were transplanted into sublethally irradiated (2.2 Gy) NSG mice ages 6–8 weeks via a tail vein injection. We used Clinical Severity Score (CSS) to evaluate the clinical status of the recipient mice[58]. The recipient mice were euthanized when they achieved high-grade CSS score during the observation period. Euthanasia was performed with cervical dislocation after sevoflurane anesthesia as per protocol. The animal experiment plan for this study has been approved in Kyushu University (A23-291-0).

### Antibodies and cell staining
Eight weeks after transplantation, spleen, liver, and bone marrow suspensions were prepared from the mice to assess the engraftment of human cells. Briefly, for FACS analysis, cells were stained with

anti-human CD45 (HI30, BioLegend, 1:50), anti-mouse CD45 (30-F11, BioLegend, 1:50), and anti-Ter119 (TER-119, BioLegend, 1:50). The cells were further stained with anti-human CD3 (UCHT1, BioLegend, 1:50), anti-human CD4 (RPA-T4, BioLegend, 1:50), anti-human CD8a (RPA-T8, BioLegend, 1:50), anti-human CD10 (HI10a, BioLegend, 1:50), anti-human CD19 (HIB19, BioLegend, 1:50), anti-human CD20 (2H7, BioLegend, 1:50), anti-human CD27 (M-T271, BioLegend, 1:50), anti-human CD34 (8G12, BD Biosciences, 1:50), anti-human CD35 (E11, BioLegend, 1:50), anti-CD38 (HIT2, BioLegend, 1:50), anti-human CD185 (CXCR5) (J252D4, BioLegend, 1:20), anti-human CD192 (CCR2) (K036C2, BioLegend, 1:20), and anti-human CD279(PD-1) (EH12.2H7, BioLegend, 1:20). Non-viable cells were excluded by propidium iodide (PI) staining. After staining, the cells were analyzed using a BD FACS Aria II (BD Biosciences), BD FACS Aria IIIu (BD Biosciences), or Attune NxT Flow Cytometer (Thermo Fisher Scientific).

For intracellular staining, cells were prepared using fixation buffer (BD Biosciences) and Perm buffer III (BD Biosciences), according to the manufacturer's instructions. Briefly, cells were stained with antibodies against surface molecules for 20 min on ice and then washed. After staining of surface molecules, cells were fixed with pre-warmed fixation buffer for 10 min and then washed. Subsequently, the cells were permeabilized with Perm buffer III for 30 min on ice and then washed. After permeabilization, the cells were stained with a PE- or Alexa Fluor 647-conjugated anti-human CXCL13 antibody (cat# IC801P, 1:10, R&D Systems and cat#IC8012R, 1:10, R&D Systems, respectively) and for 30 min. After staining, the cells were analyzed using a BD FACS Aria II (BD Biosciences), BD FACS Aria IIIu (BD Biosciences), or Attune NxT Flow Cytometer (Thermo Fisher Scientific).

### Gating strategy for FACS analysis
The general protocol to analyze FACS data in this study was as follows: (1) a size gate was used to remove debris and erythrocytes, (2) the propidium iodide (PI) and Ter119-negative fraction was gated to remove dead cells and mouse erythrocytes, (3) FSC-W-H and (4) SSC-W-H gating was performed to remove doublet cells, (5) human (h) CD45$^+$ fraction gating and hCD45$^-$mouse (m)CD45$^-$ fraction gating for in vivo reconstitution analysis. (6) CD3$^+$ cells and CD19$^+$ cells were evaluated within hCD45$^+$ fraction.

We then analyzed follicular helper Tcells (Tfh), peripheral helper Tcells (Tph), memory B cells, plasmablasts and follicular dendritic cells (FDC).

(Tfh and Tph) hCD3$^+$hCD4$^+$ fraction within hCD45$^+$ fraction was gated, then hPD-1$^{high}$hCXCR5$^+$ fraction (Tfh) and hPD-1$^{high}$CXCR5$^-$ fraction (Tph) were evaluated.

(memory B cells and plasmablasts) hCD10$^-$hCD27$^+$ fraction within hCD19$^+$ fraction was gated, then hCD20$^+$hCD38$^{dim}$ cells and hCD20$^-$hCD38$^{high}$ were evaluated as memory B cells and plasmablast.

(FDC) hCD35$^+$ fraction was evaluated as FDC within hCD45$^-$mCD45$^-$ fraction.

For the intracellular staining analysis, PI-staining was omitted.

### Immunohistochemical staining
Mouse spleens and livers were fixed in 10% neutral-buffered formalin and embedded in paraffin. Sections of 10 μm thickness were deparaffinized and antigens were retrieved using citrate-based buffer (pH 6) for 10 min (H3300, Vector Laboratories). Slides were incubated in blocking solution (3% bovine serum albumin + tris-buffered saline with Tween 20) for 1 h and incubated overnight with mouse anti-human CD3 (Dako, Clone F7.2.38, #M725401-2, 1:100) and mouse anti-human CD20cy (Dako, Clone L26, #IR604, 1:50). Slides were then incubated with an HRP-conjugated secondary antibody (Dako EnVision+ Dual Link System-HRP, Dako, #K4063). The slides were developed with diaminobenzidine (ImmPACT DAB Peroxidase (HRP) Substrate, Vector Laboratories, #SK-4105).

### IGH gene rearrangement analysis
Genomic DNA was extracted by Micro Kit (QIAGEN) according to the manufacturer's instructions. TaKaRa Taq™ (Takara, R001A) was used for PCR amplification. Primers used in PCR were FR3-JH (FR3; 5'-CCGAGGACACGGCCGTGTATTACTG-3' and JH; 5'-AACTGCTGAGGAG ACGGTGACC-3'). PCR settings were designed based on the previous studies[18,59]. Images were obtained by iBright FL1000 (Thermo Fisher Scientific).

### Imaging mass cytometry (IMC)
Tissue sections of 10 μm thickness from the liver and spleen of iMCD-NOS NSG2 mice were processed and stained according to the protocol of the Hyperion Imaging System provided by Fluidigm. The regions of interest detected by hematoxylin-eosin staining of slides of the liver and spleen were chosen for analysis. The protein expression levels of cells residing within the corresponding regions were quantified and visualized using a Hyperion Imaging System (Fluidigm). The following antibodies were used: anti-human CD4 (EPR6855, cat# 3156033D, Fluidigm, $^{156}$Gd, 1:100), anti-human CD8a (C8/144B, cat# 3162034D, Fluidigm, $^{162}$Dy, 1:50) and anti-human CD20 (H1, cat# 3161029D, Fluidigm, $^{161}$Dy, 1:200). Images of slides were prepared from PDX mice with iMCD NSG2. We obtained data of IMC from at least three independent experiments. In brief, IMC imaging mcd.files were exported into tiff files using the R software and MCD viewer (Fluidigm).

### Measurement of cytokine/chemokine and immunoglobulin protein
The mouse serums obtained from xenogeneic transplantation were assayed for 27 cytokines using the Bio-Plex Suspension Array System with Bio-Plex Pro Human Cytokine Screening 27-Plex Panel (Bio-Rad Laboratories Inc., CA, USA). The assay was performed for the following cytokines: IL-1β, IL-1Ra, IL-2, IL-4, IL-5, IL-6, IL-7, IL-9, IL-10, IL-12p70, IL-13, IL-15, IL-17, Eotaxin, FGF basic, GM-CSF, IFN-γ, IP-10, MIP-1a, PDGF-BB, MIP-1b, RANTES, TNF-alpha, and CXCL13, which were measured using a commercially available specific ELISA kit (ab269370, Abcam) according to the manufacturer's instructions. The values of each cytokine were used to generate heat maps. Human IgG, IgA, IgE and mouse albumin levels in mouse serum were measured using a commercially available ELISA kit (ab195215, ab196263, ab195216, and ab108792, Abcam) according to the manufacturer's instructions.

### In vivo therapeutic experiments using iMCD-NOS NSG mice
Female NSG mice, 6–8 weeks old, were irradiated with 2.2 Gy microwave radiation. LN cells obtained from the untreated patients with iMCD-NOS (iMCD-NOS10 and 11) were xenotransplanted into NSG mice via tail vein injection. The next day, weekly intraperitoneal injection of CXCL13-neutralizing mouse IgG antibody (15 μg/mice, BioLegend, cat#A15151D), control-mouse IgG antibody (15 μg/mice, BioLegend, MOPC-21), or daily intraperitoneal injection of rapamycin (40 μg/mice, KOM, AG-CN2-0025-M005) were initiated, respectively. Recipient mice were treated for 4 weeks.

For the evaluation of the therapeutic effect of anti-hCXCL13 antibody in the tocilizumab-refractory case (iMCD-NOS12), NSG mice were treated weekly with an intraperitoneal injection of interleukin-6 receptor antibody (tocilizumab, Selleck) (100 μg/mice), an intraperitoneal injection of anti-hCXCL13 mouse antibody (15 μg/mice) or control-mouse IgG antibody (15 μg/mice) for 6 weeks.

### Illustrations
All illustrations were generated with a license to BioRender (https://www.biorender.com).

### Statistical analysis
Statistical analyses were performed using the R software (version 4.0.3) and GraphPad Prism (version 9.5.0). Comparisons of continuous

variables between the two groups were analyzed with two-tailed unpaired *t* test. For comparisons of continuous variables between three groups, one-way ANOVA with Tukey's post-hoc tests are performed. Survival analysis was performed with log-rank test. All analysis was performed through publicly available R packages: gplots v3.1.1, dplyr v1.0.2, ggplot2 v3.3.3, readxl v1.3.1, survival v3.2-7, survminer v0.4.8. Statistical tests are indicated in the legends or corresponding method sections. Differences were considered to be statistically significant at $P \le 0.05$.

## Reporting summary

Further information on research design is available in the Nature Portfolio Reporting Summary linked to this article.

## Data availability

The original images have been deposited in Mendeley Data, V1, under the https://doi.org/10.17632/y6gtwwdsfx.2 https://data.mendeley.com/datasets/y6gtwwdsfx/2. Source data are provided with this paper.

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

## Acknowledgements

This study was supported in part by a Grant-in-Aid for Scientific Research (A) to K.A. (No. 21407314), and a Grant-in-Aid for Scientific Research (B) to T.M. (No. 21385681) and to Y.K. (No. 22494899), from the Ministry of Education, Culture, Sports, Science and Technology of Japan. This study was also supported in part by a Grant-in-Aid for the Shinnihon Foundation of Advanced Medical Treatment Research to Y.K. All figures were created with Biorender.com. All illustrations were generated with a license to BioRender.

## Author contributions

T.H., Y.K., T.M., H.N., K.A., and K.Y. designed the research. T.H., Y.K., H.N., and K.U. performed research and collected data. T.H., Y.K., T.M., H.N., K.A., A.K., T.K., and K.Y. analyzed and interpreted data. T.H., Y.K., K.A., and K.Y. wrote the manuscript.

## Competing interests

T.H., Y.K., H.N., K.A., and K.Y. have filed a patent application based on this work. The remaining authors declare no competing interests.
