## [Peer Review File · Nature Communications]

Peripheral helper-T-cell-derived CXCL13 is a crucial pathogenic factor in idiopathic multicentric Castleman diseaseREVIEWER COMMENTS

Reviewer #1 (Remarks to the Author):

The Manuscript from Harada et al summarises the findings of a reverse translational approach aimed at identifying, through xenotransplant models, the factors contributing to poor outcome in idiopathic multicentric Castelman disease.

The results suggest that blocking CXCL13, a chemokine previously identified as increased in the serum of these patients, prevents the severe inflammation observed in the model. The authors therefore imply that blocking CXCL13 may be beneficial in patients affected by this very severe condition.

The approach is original and extremely informative, the experimental plan is well thought of. Some of the conclusion are slightly beyond the data shown as for the conclusions and few points would deserve clarification.

1. The authors should describe the observations that lead to define "systemic inflammation". While a cachectic appearance of mice and some hair loss on the dorsal skin, may be suggestive of inflammation, an objective measure would be more informative.

2. Lethal inflammation: some times the authors state that mice succumbed (in a period of time ranging from few days to several weeks) , but in the methods they state that mice were euthanised if showing "frailty". The actual outcome of the mice should be described in detail and reflected in the graphs.

3. Figure 1D, as well as all the other Immunohistochemistry or immunofluorescence panels , do not show any anti Mouse control or negative control. Magnification reference is also missing.

4. the N described in the methods does not match the dots shown in figure 2B and 3B-D.

5. The experiments in figure 4 adopt a different control than the others, and CD34+ HSPCs appear for the first time here. The authors should justify this choice compared to the controls adopted in the first part of the study.

Within the limitation mentioned above, this is an interesting study, which has a high translational potential. Nevertheless the authors should limit their discussion to the observations in the animal model, and perhaps also comment on the variability observed across patient donors.

Reviewer #2 (Remarks to the Author):

Comments to authors:

The investigators appear to have developed a mouse model for idiopathic multicentric Castleman disease - a deadly rare disease with poor understanding of its disease pathogenesis. The field is in need of both a reliable model system and more effective treatments, both of which are investigated in this paper. I applaud the authors for this excellent and important work.

In exploring their model, the investigators report a role for expanded plasmablasts, expanded TPH cells producing CXCL13, and reveal that CXCL13 blockade can dramatically protect mice in their model. This is potentially extremely helpful to the field, but evaluation of the model is hampered by a quite limited characterization and supporting data of it or the control groups it is compared to. The donor patients are not very well characterized. The phenotype of the mice, including lymph node size, is very limited. For some reason, the authors chose to provide "representational" images rather than comparisons between groups with replicates or even with controls. These between group comparisons with replicates need to be performed and shared.

Further, the controls change between experiments without explanation for why they are changing. It will be important for the authors to explain why the controls have changed. Are CD34+ HSCs a good control for comparing against lymph node tissue from iMCD patients? One would suspect that there would be a striking difference just based on cell types even if given from the same donor. The data supporting TPH expansion or their role as the cellular source of CXCL13 is particularly limited. I look forward to reviewing a revised version of this manuscript where the authors have more clearly described the phenotypes, considered alternative control groups, performed comparisons between groups, and demonstrated that their corollary insights into disease

pathogenesis are reproducible. See below for a listing of specific revisions recommended:

Abstract

Line 42:

States: "Immunodeficient mice transplanted with lymph node (LN) cells from iMCD patients exhibited iMCD-like lethal inflammation"

It seems this sentence needs to be completed with a description of what the comparison group was, like "compared to mice transplanted with cells from non-iMCD control lymph nodes who did not exhibit inflammation."

Line 43

States: "Depletion of human CD3+ T cells from grafts failed to induce lethal inflammation in vivo." Shouldn't it say also which cell type depletion did not rescue the mice like:

"Depletion of human CD3+ T cells from grafts prevented the lethal inflammation phenotype in vivo; depletion of X cell types did not rescue the mice."

Line 48: The authors mention "iMCD-NOS" before saying what iMCD-NOS is. Consider removing or explaining iMCD-NOS in the abstract.

Line 48: I think the authors should consider changing "indicating" to "suggesting"

Introduction:

Line 92:

I think the authors should consider changing "etiology" to "pathogenesis"

Line 136-140:

The mice injected with LN cells showed weight loss/vascularity, but was there any characterization of the lymph nodes? Did the mice show histological features seen in iMCD? Were they enlarged? It would be helpful if the authors more clearly demonstrate that human disease is recapitulated in the mice.

Line 159:

It is not clear why the investigators have "data not shown" are not sharing the results of the immunoglobulin heavy chain rearrangement studies. They should also prove sensitivity of their approach by doing this in control mice that were injected with LN cells from lymphoma patients.

A general suggestion to avoid this confusion is that everywhere it says "data not shown", the data should be shown in the supplemental materials.

Line 174:

Comparing the conclusion here to figure 2b, it seems there was not a significant increase in %cd45+ cells in spleen injected from P5. Seems to be a good amount of heterogeneity in results. Not only in this figure, but others as well. Would like to see the authors comment on this in the discussion/results section.

Line 192:

States "but not in humanized NSG mice that reconstituted normal human hematopoiesis by xenotransplantation of cord blood (CB)-derived CD34+ hematopoietic stem/progenitor cells (HSPCs)"

what about the mice who got the lymphoma and reactive lymph node transplant? Was there expansion of these cells?

Line 210:

States: "and NSG mice reconstituted with human normal hematopoiesis (n=3) after transplantation of CB-derived CD34+ HSPCs."

What about with mice with reactive or lymphoma infusion like the previous ones?

Line 220:

States:

In iMCD-NOS NSG2 mice, IFN- γ , GM-CSF, CXCL13, and TNF- α levels were dramatically elevated compared to those in the serum of patient 2 (Fig. 4f).

were these elevated in the human patient compared to healthy controls when the patient was in active disease before the transplant?

Line 238:

Should "and an intraperitoneal" should be "or an intraperitoneal"?

Line 278 states "our study clearly showed that iMCD-NOS depends on immune cell crosstalk rather than cell-autonomous abnormalities of specific lineage cells."

Have the authors considered that this could also just indicate that T cells are the cell-autonomous problem?

Line 338:

These malignant DLBCL cells did not graft in the body? and cause problems?

Line 359:

The investigators indicate that LN cells were injected in a FBS-containing medium. This is a potentially concerning technical issue that can itself induce cytokines and other unwanted inflammatory responses. Did the authors repeat this experiment by injecting cells in a serum-free isotonic solution (e.g. PBS or RPMI) to confirm whether they still observe an iMCD LN-specific lethality phenotype? If not, I think this should be strongly considered.

Lines 431-433:

It states: "Statistical tests were performed in R and for more details you could consult the methods/figure legends." However, I did not see specific explanations of the stat tests used in any analysis. I strongly encourage the authors to report these statistical tests and explanations for them more clearly.

Further, I strongly encourage the authors to replace all "representational" images in figures with statistical comparisons between groups with replicates.

Line 455:

Figure 1 potentially achieves a very exciting milestone in identifying a way to model Castleman Disease in mice, but the characterization of the patients and mice is very limited. Investigators report that mice became cachectic, but no weights are given. This should be provided with statistical tests and error bars. In addition to weights, repeated blood sampling would be helpful at intermediate timepoints to look for biochemical evidence of inflammation (blood count abnormalities, elevated quantitative immunoglobulins, elevated LDH, CRP, ESR, IL6) consistent with what is seen in iMCD. The authors also need to report the size of lymph nodes in these mice and compare them to controls as this is a major criterion of diagnosis of the disease in humans. Lymph node histology, another hallmark of diagnosis, is also missing to see whether this recapitulates human disease. The survival curve includes wide variability but no work is done to correlate this to the patient's clinical characteristics – was there variability in the patients' clinical phenotypes before transplant?

The survival curve mentions a day 70 endpoint in which the iMCD-NOS NSG3 mice were euthanized for "frailty" (page 5, line 139). Can the authors comment about whether all mice from iMCD-NOS NSG groups were euthanized for a similar reason with frailty? Can you define this criterion further?

H&E of spleen and liver from a single mouse is not clearly representative and should be accompanied by quantitative assessment of CD3+ and CD20+ cell infiltration into these organs in all iMCD and control LN-injected mice.

Fig. 1c, would have liked to see some quantification of weight loss over the course of experimentation, rather than representative images here. Also, it is not mentioned if there was

any other clinical evidence of systemic inflammation besides weight loss/vascularity in the ear, were inflammatory markers measured?

Line 464: did you look at LN size and histology?

Line 473:

Figure 2 reports splenic plasmablast expansion in all iMCD LN recipients, but these data are not provided in a summary figure with statistical tests to demonstrate this expansion. Only a "representative" flow from a single mouse is shown. It is unclear whether this is seen across mice with the same LN donor, different iMCD LN donors, or control LN donors. The timepoint post-injection is also not given. Can the authors add these data to the paper?

Though not shown in the flow plots, the text (page 5, line 156) reports the plasmablasts were CD20+ though plasmablasts are typically negative for CD20 (Ellebedy et al. *Nat Immunol.* 2016.; Fink et al. *Front. Immunol.* 2012.; Covens et al. *Blood.* 2013.; Owczarczyk et al. *Sci. Trans. Med.* 2011.) Can the authors comment on this in the discussion?

A new donor - iMCD-NOS NSG4 - is introduced here for unclear reasons. Can the authors explain this? It may be because the investigators ran out of LN cells from the other donors, but it is not clear from the text why this was done.

The authors should have considered performing blood sampling for quantitative immunoglobulins for a broader range of donors, not just one at this unclear timepoint. IgE was assessed for unclear reasons and IgM was not measured.

Line 473:

New patient (iMCD-NOS NSG4) introduced in this figure. Not mentioned in Fig.1 (although present in Supp. Table 1). Can the authors provide more information about this patient (and all patients) who provided lymph nodes for this study?

What is the timepoint for FACS images post injection? Are these expansions of T and B cells consistent across mice/patients? This was not clear from the text/figure. Can groupwise comparisons be done?

Line 477:

States "A representative FACS plot of cells in the spleen" - can you show compared to what is seen in the control mice instead of a representative plot?

Line 495:

Figure 3 reports that T-B interaction was required for the inflammatory phenotype by performing CD3-depletion on two iMCD LN donors and assessing splenic %CD45 cells and sera immunoglobulin levels. Panel A reports that the mice injected with the unmanipulated LN cells were followed for 8 weeks, but this donor seemed to induce much faster lethality in Figure 1b. Panel A is a cartoon that reports "cachexia" was observed in the unmanipulated iMCD LN recipients, but no data (e.g. weight loss, survival) is given to support the schematic. Mice who received P5 +/- CD3 depletion did not appear to have a significant difference in splenic %CD45 as the authors state in the text (page 6, line 174-5). It is also unclear how the close proximity of T/B cells in the spleen and liver shown on the immunofluorescent microscopy proves any T/B interaction as described in the text. B cells and T cells often border one another in normal lymph nodes. CD8 staining in blue is difficult to see.

Line 501:

Figure 3B legend. Does this mean that none of the immune cells grafted?

Line 512:

Figure 4 reports that Tph population is specifically expanded in iMCD and are the cellular source of CXCL13. Altogether, a larger flow panel that includes TIGIT, ICOS, CCR, and CXCL13 (with ex vivo CD3/28 re-stimulation) should be considered to characterize the Tph cells across all iMCD and control samples as had been done to better characterize these cells in other diseases where they

have been implicated (Rao et al. Nature. 2017.; Uzzan et al. Nature Medicine. 2022).

Panel A should include percentages in gate. Panel b description (line 517 - 520) does not include which statistical tests (ANOVA?) were done: Only a simple p value is given. The color key in panel 2 is illegible.

The use of only representative flow plots for CCR2 (panel c) and CXCL13 (panel g) expression is particularly unusual and should be shown for all iMCD and controls samples as groupwise comparisons with statistical tests done. TPH are certainly an excellent candidate as the cellular source of CXCL13, which has been shown to be dramatically increased in plasma of iMCD patients, but a very slight shift in the peak of one representative flow plot without showing whether it is true across samples or whether it is increased compared to control is insufficient to reach this conclusion. Furthermore, other candidates for the cellular source of iMCD (such as follicular dendritic cells) are not addressed.

I was not able to follow why the investigators switch between spleen in the recipient mice and LN/PB characterization in the human donor even though they have only previously characterized the recipients spleens to this point. I would recommend one set of data dedicated to high-parameter flow immunophenotyping of the donor LN cells from iMCD and control LNs and another focused on flow immunophenotyping of recipient NSG mouse LN and spleen. Altogether, an orthogonal approach, such as scRNAseq and/or IF would also make sense to prove expansion of these cells in iMCD.

The control that was used in panel b is confusing to me: Why are cord blood CD34+ HSPCs used as controls? It is not clear why control LN injected recipients (the controls used elsewhere in the paper) were not used here. SLE/RA controls would also be nice if the investigators are trying to prove specificity as they state in the figure legend title (line 514). The use of cord-blood controls instead of recipient mice injected with control LN cells is also unusual for the cytokine comparison.

Surprisingly, despite being injected with LN cells from a single donor, the mice seem to have widely different relative expression of cytokines (even CXCL13 appears down in one mouse) and no single chemokine appears to be increased in every mouse even though they are presumably all technical replicates from a single donor. Can the authors comment on this?

I am also unclear on how comparing change in chemokines/cytokines from patient sera and the recipient mice as done in panel f helps to implicate molecules involved in disease pathogenesis: Shouldn't CXCL13 be high in the patient too since this was previously shown (Pierson. Am J Hematol. 2018.)? Where is patient 2 in therapy course (what treatments were given) / status of disease activity when sera was collected? At diagnosis/during flare? Already treated and responding? In text (page 7, line 220- 222), the authors argue that TPH cells had been shown to produce all of these cytokines in other contexts - they should consider doing an ex vivo restimulation to show whether it occurs in their model.

Line 566:

Figure 5 makes the case that CXCL13 blockade can improve survival in their mouse model of iMCD. This figure is extraordinarily promising and encouraging to the field. However, I would recommend that IL6 blockade be included as this is accepted as first-line therapy for iMCD (van Rhee et al. Blood. 2018). It would also therefore be important to know whether the patients were treated with IL6 blockade and whether they responded to see how well the mouse model recapitulates the patient's disease.

Supplemental Table:

Clinical data provided is insufficient to understand disease severity for each patient and indeed is insufficient to establish the diagnosis as many patients included seem to not satisfy minor laboratory criteria to diagnose the disease (Fajgenbaum et al. Blood. 2017.) How the patients were treated and whether they responded would also be useful. I would also strongly recommend inclusion of representative H&E images of patient lymph nodes to prove correctness of diagnosis as there is a high degree of pathologic discordance.

Reviewer #3 (Remarks to the Author):

In this study, Harada and colleagues established a mouse model for iMCD-NOS by engrafting LN cells from patient into immunodeficient mice. After the engraftment, the recipient mice exhibited iMCD-like fatal systemic inflammation, recapitulating the human disease. This humanized mouse model is featured by high level of human antibodies as expansion of hCD4+hPD-1^{high}hCXCR5-hCCR2⁺ 187 Tph cells . Moreover, the authors could show that levels of hCXCL13 and some other cytokines including hIL-6 were significantly higher in mice engrafted with LN cells from patients than controls, and neutralizing hCXCL13 with anti-hCXCL13 antibody improved the survival of iMCD-NOS mice. Based on the above observations, the authors conclude that Tph cells that produce CXCL13 play a critical role in the pathogenesis of iMCD. Findings from this novel humanized mouse model shed new lights to the pathogenesis of iMCD-NOS. However, this study has several major limitations.

1. As the authors mentioned in the introduction section, a major question raised from the clinical side is that approximately half of iMCD patients do not respond to IL-6 inhibition, which indicating alternative biological pathways are involved in the pathogenesis of iMCD-NOS. A humanized mouse model provides an opportunity to address this issue. However, the study does not provide convincing evidence in this direction. Therefore, this study is not of great significance to the field in this regard.

2. Mice engrafted with LN cells from iMCD-NOD patients showed body weight loss, depilation , inflammation and death, which are also features of graft-versus-host disease (GVHD). Since GVHD is common and major complication that occurs following engrafting human immune cells into immunodeficient mice (PMID: 35993192), the authors need to provide evidence for the lack of GVHD in this humanized mouse model.

3. The authors claimed that transfer of LN cells from iMCD patients leads to the development of lethal systemic inflammation in mice. However, in figure 1, only representative histology of lung and livers are presented. To support their statement, evaluation the histology of other organs is required, and statistical analysis need to be performed to show the difference between the two groups of mice. Due to the deficiency of T and B cells, the spleen of immunodeficient mice is tiny and featured by the lack of germinal center, and transfer of human immune cells into those mice can restore the GC (PMID: 34248959). It is interesting to know whether engraftment of LN cells from iMCD patients and controls restore the GC in the murine spleen.

4. Figure 2a shows only a representative FACS plot of a mouse. Data from more mice engrafted with iMCD and control LN cells are required for quantitative statistics. In figure 2b and 2c, only mice engrafted LN cells from one patients (P4) were used for statistical analysis, which is not convincing.

5. This study claims that Tph cells that produce CXCL13 play a critical role in the pathogenesis of iMCD. However, the author did not provide evidence that the Tph cells are the major source of hCXCL13.

6. In figure 4, the difference in hCD4+hPD-1^{high}hCXCR5-hCCR2⁺ 187 Tph cells and cytokines between the disease group and controls are from the comparison between mice engrafted with LN cells from iMCD patients and mice engrafted with CB-derived CD34⁺ HSPCs. From the reviewer's point of view, the control group should be mice engrafted with LN cells from patients with other disease or healthy subjects.

7. In figure 5, only survival data are presented. It is interesting to know the effect of blocking CXCL13 on other immunological, histological phenotypes.

We thank the editor and the reviewers for positive comments and their valuable feedback. We have responded to the reviewers' concerns in a point-by-point manner and have accordingly revised our manuscript. We believe the addition of the new data and experiments suggested by the reviewers improves substantially the impact of our work.

REVIEWER COMMENTS

Reviewer #1 (Remarks to the Author):

The Manuscript from Harada et al summarises the findings of a reverse translational approach aimed at identifying, through xenotransplant models, the factors contributing to poor outcome in idiopathic multicentric Castelman disease.

The results suggest that blocking CXCL13, a chemokine previously identified as increased in the serum of these patients, prevents the severe inflammation observed in the model. The authors therefore imply that blocking CXCL13 may be beneficial in patients affected by this very severe condition.

The approach is original and extremely informative, the experimental plan is well thought of. Some of the conclusion are slightly beyond the data shown as for the conclusions and few points would deserve clarification.

1. The authors should describe the observations that lead to define "systemic inflammation". While a cachectic appearance of mice and some hair loss on the dorsal skin, may be suggestive of inflammation, an objective measure would be more informative.

Response

We appreciate the reviewer's helpful comment. To evaluate the systemic inflammation of the recipient mice in an objective manner, we expanded the analysis of body weight changes between iMCD-NOS mice and control mice. As shown in the new Fig2c, iMCD-NOS NSG mice exhibited significant body weight loss as compared to control mice.

2.Lethal inflammation: some times the authors state that mice succumbed (in a period of time ranging from few days to several weeks) , but in the methods they state that mice were euthanised if showing "frailty". The actual outcome of the mice should be described in detail and reflected in the graphs.

Response

We appreciate your helpful suggestion. The actual outcomes of the iMCD-NOS NSG mice shown in Fig1b are summarized in the new Supplementary Fig.2a. We used Clinical Severity Score (CSS) to evaluate the clinical status of the recipient mice (PMID 21737102). The recipient mice were sacrificed when they achieved high-grade CSS score during the observation period. New Supplementary Table 2 summarized the detailed information of xenotransplantation experiments including CSS score and frequencies of engrafted human cells. We also described the employment of CSS in the section of methods (line 434-437).

3. Figure 1D, as well as all the other Immunohistochemistry or immunofluorescence panels , do not show any anti Mouse control or negative control. Magnification reference is also missing.

Response

Related to your question 5, the LN cells from control subjects failed in the engraftment and expansion of human blood cells *in vivo* (new Fig.1d and Supplementary Table 2). We, therefore, omitted the histological data of negative control mice. According to your helpful suggestion, we showed the HE staining and immunohistochemistry analysis of the spleen and liver from NSG mice without transplantation as negative controls in the new Fig1e. The magnification reference is also shown in the new Fig1e.

4.the N described in the methods does not match the dots shown in figure 2B and 3B-D.

Response

We appreciate your critical comments. We corrected the data of new Fig.2c and 3b-d. New Supplementary Table 2 summarized the detailed information of xenotransplantation experiments.

5. The experiments in figure 4 adopt a different control than the others, and CD34+ HSPCs appear for the first time here. The authors should justify this choice compared to the controls adopted in the first part of the study.

Within the limitation mentioned above, this is an interesting study, which has a high translational potential. Nevertheless the authors should limit their discussion to the observations in the animal model, and perhaps also comment on the variability observed across patient donors.

Response

We highly appreciate your positive and important comments. LN cells from control

subjects failed in the efficient engraftment and reconstitution of human blood cells in the recipient mice. We, therefore, employed the NSG mice reconstituted with normal human hematopoiesis via xenotransplantation of human CD34⁺HSPCs as control subjects. Furthermore, we extended xenotransplantation experiments using reactive LN cells (new control 6 and 7) in the revised manuscript, and again confirmed that most of reactive LN cells (control 1, 6, and 7) and LN cells from lymphoma samples (control 2, 3, 4, and 5) failed in the efficient engraftment of human cells in the recipient mice (summarized in new Supplementary Table 2 and new Fig.1d). According to the reviewer's helpful suggestions, we modified the results section to describe the reason why we employed the NSG mice transplanted with CD34⁺HSPCs as control subjects in Fig.4 (line 210-214). We also added the sentences to describe the limitation of PDX model in the revised manuscript (line 363-379) and modified Fig.6 and its title to accurately summarize the findings observed in our PDX model (line 763-766). We highly appreciate your constructive comments.

Reviewer #2 (Remarks to the Author):

Comments to authors:

The investigators appear to have developed a mouse model for idiopathic multicentric Castleman disease - a deadly rare disease with poor understanding of its disease pathogenesis. The field is in need of both a reliable model system and more effective treatments, both of which are investigated in this paper. I applaud the authors for this excellent and important work.

In exploring their model, the investigators report a role for expanded plasmablasts, expanded TPH cells producing CXCL13, and reveal that CXCL13 blockade can dramatically protect mice in their model. This is potentially extremely helpful to the field, but evaluation of the model is hampered by a quite limited characterization and supporting data of it or the control groups it is compared to. The donor patients are not very well characterized. The phenotype of the mice, including lymph node size, is very limited. For some reason, the authors chose to provide "representational" images rather than comparisons between groups with replicates or even with controls. These between group comparisons with replicates need to be performed and shared.

Further, the controls change between experiments without explanation for why they are changing. It will be important for the authors to explain why the controls have changed. Are CD34+ HSCs a good control for comparing against lymph node tissue from iMCD patients? One would suspect that there would be a striking difference just based on cell types even if given from the same donor. The data supporting TPH expansion or their role as the cellular source of CXCL13 is particularly limited. I look forward to reviewing a revised version of this manuscript where the authors have more clearly described the phenotypes, considered alternative control groups, performed comparisons between groups, and demonstrated that their corollary insights into disease pathogenesis are reproducible. See below for a listing of specific revisions recommended:

Response

We highly appreciate your positive comments and constructive suggestions. Regarding the control subjects, we employed CD34⁺HSPCs for the control xenotransplantation experiments, because non-iMCD LN cells failed in the efficient reconstitution of human hematopoiesis in NSG mice. We extended xenotransplantation experiments using reactive LN cells (new control 6 and 7) in the revised manuscript, and again confirmed that reactive LN cells (control1,6, and 7) as well as LN cells from lymphoma samples

(control 2,3,4, and 5) failed in the efficient engraftment of human cells in the recipient mice (summarized in new supplementary Table2 and new Fig.1d). We modified the results section to describe the reason why we employed the NSG mice transplanted with CD34⁺HSPCs as control subjects in Fig.4 (line 210-214).

Regarding the source of human CXCL13, we added the several data in the revised manuscript. First, we showed that human follicular dendritic cells (FDCs) and Tfh cells which could secrete CXCL13 were almost undetectable in the iMCD-NOS NSG mice (new Supplementary Fig. 5c). We, therefore, focused on the expanded human Tph cells as the source of human CXCL13 at least in our xenograft model. Second, we confirmed the increased frequencies of human Tph cells in the primary LN samples obtained from the patients with iMCD-NOS (new Fig. 4d). Third, we confirmed hCXCL13 expression in human Tph cells in iMCD-NOS NSG mice (new Supplementary Fig.6b). These extended data support our hypothesis that Tph cells can be a source of CXCL13 production in iMCD-NOS. Of course, these data do not exclude the possibility that FDCs and Tfh cells as the cellular source of CXCL13 in the patients with iMCD-NOS. We, therefore, described the limitation of our study regarding the source of CXCL13 in iMCD-NOS (line 375-379). We also modified Fig.6 and its title to accurately show the findings observed in our PDX model (line 763-766).

Abstract

Line 42:

States: "Immunodeficient mice transplanted with lymph node (LN) cells from iMCD patients exhibited iMCD-like lethal inflammation"

It seems this sentence needs to be completed with a description of what the comparison group was, like "compared to mice transplanted with cells from non-iMCD control lymph nodes who did not exhibit inflammation."

Response

According to the reviewer's comment, we modified the abstract as follows "Immunodeficient mice transplanted with lymph node (LN) cells from iMCD patients exhibited iMCD-like lethal inflammation compared to mice transplanted with LN cells from non-iMCD patients without inflammation" (line 40-42) to describe the control groups. We appreciate your helpful suggestion.

Line 43

States: “Depletion of human CD3+ T cells from grafts failed to induce lethal inflammation in vivo.”

Shouldn't it say also which cell type depletion did not rescue the mice like:

“Depletion of human CD3+ T cells from grafts prevented the lethal inflammation phenotype in vivo; depletion of X cell types did not rescue the mice.”

Response

We highly appreciate the reviewer's helpful comment. As you mentioned, we should try B cell depletion in mice; however, we could not evaluate the effect of B-cell depletion because of the limited cell number and clinical samples available in the current study. Thank you for your constructive suggestion.

Line 48: The authors mention “iMCD-NOS” before saying what iMCD-NOS is. Consider removing or explaining iMCD-NOS in th abstract.

Response

According to the reviewer's comment, we explained iMCD-NOS in the abstract (line 38).

Line 48: I think the authors should consider changing “indicating” to “suggesting”

Response

According to the reviewer's comment, we changed “indicating” to “suggesting” in the abstract (line 48).

Introduction:

Line 92:

I think the authors should consider changing “etiology” to “pathogenesis”

Response

According to the reviewer's comment, we changed “etiology” to “pathogenesis” in the revised manuscript (line 93).

Line 136-140:

The mice injected with LN cells showed weight loss/vascularity, but was there any characterization of the lymph nodes? Did the mice show histological features seen in iMCD? Were they enlarged? It would be helpful if the authors more clearly demonstrate

that human disease is recapitulated in the mice.

Response

NSG mice completely lack mouse T/B/NK cells, and therefore, lymph nodes are rarely detected in the steady state. Even after the reconstitution of human hematopoiesis, it is hard to detect lymphadenopathy in mice. Instead of lymphadenopathy, splenomegaly was observed in the iMCD-NOS NSG mice. We presented these data to support our hypothesis that xenotransplantation of LN cells from iMCD-NOS patients could recapitulate iMCD-NOS -like disease status *in vivo* (new Supplementary Fig.3b and 3c) (line 178-180). We highly appreciate your helpful comments.

Line 159:

It is not clear why the investigators have “data not shown” are not sharing the results of the immunoglobulin heavy chain rearrangement studies. They should also prove sensitivity of their approach by doing this in control mice that were injected with LN cells from lymphoma patients.

A general suggestion to avoid this confusion is that everywhere it says “data not shown”, the data should be shown in the supplemental materials.

Response

According to the reviewer’s helpful suggestion, we presented the data of immunoglobulin heavy chain rearrangement analysis of human B cells in the iMCD-NOS NSG mice (new Supplementary Fig.3a) (line 170). Regarding the control subjects, the engraftment of control LN cells was rarely observed in the recipient mice as we mentioned above (new Supplementary Table 2 and new Fig.1d), so we could not perform the immunoglobulin heavy chain rearrangement analysis in the control subjects. Normal CD34⁺ HSPCs always give rise to polyclonal B cell development in NSG mice as we previously reported (PMID 21840488).

Line 174:

Comparing the conclusion here to figure 2b, it seems there was not a significant increase in %cd45⁺ cells in spleen injected from P5. Seems to be a good amount of heterogeneity in results. Not only in this figure, but others as well. Would like to see the authors comment on this in the discussion/results section.

Response

We appreciate your constructive comment. As we mentioned above (please refer to the response to question 4 from Reviwer#1), we corrected the data of Fig.2b and 3b-d (new Fig.2c and 3b-d). Therefore, there was a significant increase in %CD45⁺ cells in the

spleen injected from P5. However, it is true that experimental results have varied. Hence, we added the new table describing the chimerism of human hematopoietic cells in the iMCD-NOS NSG and control NSG mice (new Supplementary Table2). As shown in this table, the reconstitution of human CD45⁺ cells varied among individual patients. As you suggested, such kinds of differences in the reconstitution or survival of the recipient mice might reflect the clinical features of the original patients. We discussed this point in the discussion (line 363-368). We also added the data of CHAP score (PMID 28880697) in Supplementary Table1 to show the disease activity of the individual patients (line 412-413).

Line 192:

States “but not in humanized NSG mice that reconstituted normal human hematopoiesis by xenotransplantation of cord blood (CB)-derived CD34⁺ hematopoietic stem/progenitor cells (HSPCs)”

what about the mice who got the lymphoma and reactive lymph node transplant? Was there expansion of these cells?

Response

We appreciate your important comments. As we mentioned above, LN cells from control subjects failed in the efficient reconstitution of human hematopoietic cells in vivo (new Supplementary Table 2 and new Fig1.d). We, therefore, compared the frequencies of human Tph cells between iMCD-NOS NSG mice and NSG mice reconstituted with normal human hematopoiesis using CD34⁺ HSPCs. We modified the results section to describe the reason why we employed the NSG mice transplanted with CD34⁺HSPCs as control subjects in Fig.4 (line 210-214).

Line 210:

States: “and NSG mice reconstituted with human normal hematopoiesis (n=3) after transplantation of CB-derived CD34⁺ HSPCs.”

What about with mice with reactive or lymphoma infusion like the previous ones?

Response

We could not evaluate the chemokine/cytokine profiling of the NSG mice transplanted with non-iMCD-NOS LN cells due to the lack of engraftment of human cells.

Line 220:

States:

In iMCD-NOS NSG2 mice, IFN- γ , GM-CSF, CXCL13, and TNF- α levels were dramatically elevated compared to those in the serum of patient 2 (Fig. 4f).

were these elevated in the human patient compared to healthy controls when the patient was in active disease before the transplant?

Response

Thank you for your constructive comments. We confirmed the elevated IFN- γ , GM-CSF, CXCL13 and TNF- α in the patient (P2) serum compared to those in the sera of healthy donors (n=3) (new Supplementary Fig.5b) (line 245-247).

Line 238:

Should “and an intraperitoneal” should be “or an intraperitoneal”?

Response

Thank you for your helpful comment. We agreed with this point and corrected the sentence in the revised manuscript (line 266).

Line 278 states “our study clearly showed that iMCD-NOS depends on immune cell crosstalk rather than cell-autonomous abnormalities of specific lineage cells.”

Have the authors considered that this could also just indicate that T cells are the cell-autonomous problem?

Response

We appreciate your important comment. Depletion of T cells blocked the development of iMCD-like disease status *in vivo*. We, therefore, considered that different from cell-autonomous expansion of B cells and T cells like leukemia/lymphoma, iMCD-NOS should be an immunological disorder in which immune cell cross talk should play a critical role. To improve the understanding of the reader, we changed the sentence as follows; “our study indicated that the pathogenesis of iMCD-NOS depends on the abnormal activation of chemokine/cytokine-mediated immune cell crosstalk” (line 314-316).

Line 338:

These malignant DLBCL cells did not graft in the body? and cause problems?

Response

Yes. As we discussed above, control LN cells from non-iMCD-NOS did not show efficient engraftment in the recipient mice as shown in new Fig.1d and new Supplementary Table 2. Past reports have indicated that the creation of PDX mice for DLBCL is technically difficult, and that specific transplantation methods and genetic modification of the mice are necessary (PMID: 26773040).

Line 359:

The investigators indicate that LN cells were injected in a FBS-containing medium. This is a potentially concerning technical issue that can itself induce cytokines and other unwanted inflammatory responses. Did the authors repeat this experiment by injecting cells in a serum-free isotonic solution (e.g. PBS or RPMI) to confirm whether they still observe an iMCD LN-specific lethality phenotype? If not, I think this should be strongly considered.

Response

We highly appreciate your critical comment. At the time of preparation of single cell from LN, we used RPMI supplemented with 10% FBS as described in the methods. After that, we washed LN cells with PBS, and resuspended FBS-free RPMI. So, we injected LN cells in a serum-free solution. We corrected the description of the corresponding part (line 432-434).

Lines 431-433:

It states: "Statistical tests were performed in R and for more details you could consult the methods/figure legends." However, I did not see specific explanations of the stat tests used in any analysis. I strongly encourage the authors to report these statistical tests and explanations for them more clearly.

Response

We apologize for the omission of the method description. All continuous variables were analyzed with two-tailed unpaired *t*-test. For two-group comparisons, the error bar is presented as mean \pm standard error (SE). For other data with multiple samples, the data are presented as mean \pm standard deviation (SD) to indicate variability. Data for continuous variables that include iMCD, TBI negative control and other control groups are analyzed by one-way ANOVA with Tukey's post-hoc tests for multiple comparisons. Statistical analysis methods are described in each figure legend. Statistical analysis and figure generation was performed using R v4.0.3 or GraphPad Prism (version 9.5.0). Therefore, we added following sentences in Methods section; "Comparisons of continuous variables between the two groups were analyzed with two-tailed unpaired *t*-test. For comparisons of continuous variables between three groups, one-way ANOVA with Tukey's post-hoc tests are performed. Survival analysis was performed with log-rank test. All analysis was performed through publicly available R packages: gplots

v3.1.1, dplyr v1.0.2, ggplot2 v3.3.3, readxl v1.3.1, survival v3.2-7, survminer v0.4.8" (line 519-525). Error bars represented as mean \pm SE or SD. We noted what they represent in figure legends (line 633-634, line 649, line 690-692, and line696).

Further, I strongly encourage the authors to replace all "representational" images in figures with statistical comparisons between groups with replicates.

Response

Thank you for your constructive comments. As for the representative images, we have converted them to continuous variables and performed statistical processing (new Fig.1d, Fig.2b, Supplementary Fig.3c, Supplementary Fig.4a, Fig.4d, and Supplementary Fig.5a). As we mentioned above, systemic inflammation in Fig.1c was evaluated by CSS and described in Supplementary Table 1.

Line 455:

Figure 1 potentially achieves a very exciting milestone in identifying a way to model Castleman Disease in mice, but the characterization of the patients and mice is very limited. Investigators report that mice became cachectic, but no weights are given. This should be provided with statistical tests and error bars. In addition to weights, repeated blood sampling would be helpful at intermediate timepoints to look for biochemical evidence of inflammation (blood count abnormalities, elevated quantitative immunoglobulins, elevated LDH, CRP, ESR, IL6) consistent with what is seen in iMCD. The authors also need to report the size of lymph nodes in these mice and compare them to controls as this is a major criterion of diagnosis of the disease in humans. Lymph node histology, another hallmark of diagnosis, is also missing to see whether this recapitulates human disease. The survival curve includes wide variability but no work is done to correlate this to the patient's clinical characteristics – was there variability in the patients' clinical phenotypes before transplant?

Response

We added the new data describing the body weight change in new Fig.2c. Regarding lymphadenopathy, as we responded to your question (line136-140), even after the reconstitution of human hematopoiesis, we could not obtain the enlarged LN in the recipient mice. Instead of lymphadenopathy, we could observe the enlarged spleen as compared to the control mice transplanted with non-iMCD-NOS LN cells (new Supplementary Fig.3b and 3c) (line 178-180).

Repeated blood sampling would be helpful to obtain the biomarker of inflammation; however, we considered that the repeated blood sampling might affect the survival of the recipient mice, because the mice exhibited very cachectic status as shown Fig.1 and new Fig.2c. So, we could not perform repeated blood sampling of the recipient mice transplanted with iMCD-NOS LN cells.

Regarding the variability of the survival of the recipient mice, we added the new table describing the chimerism of human hematopoietic cells in the iMCD-NOS NSG and control NSG mice to show the detailed information (new Supplementary Table 2). As you suggested, such kinds of differences in the reconstitution or survival of the recipient mice might reflect the clinical features of the original patients. We discussed this point in the discussion (line 363-368). We highly appreciate your helpful comments.

The survival curve mentions a day 70 endpoint in which the iMCD-NOS NSG3 mice were euthanized for "frailty" (page 5, line 139). Can the authors comment about whether all mice from iMCD-NOS NSG groups were euthanized for a similar reason with frailty? Can you define this criterion further?

Response

Thank you for your constructive comments. As we mentioned above, we sacrificed mice when these reached score 4 with clinical severity score (CSS). However, some mice suddenly died due to the rapid progression of inflammation. The detailed mice clinical course is shown in new supplementary Fig.2a in the form of a swimmer plot. All clinical severity scores of the mice are described in new Supplementary Table 2.

H&E of spleen and liver from a single mouse is not clearly representative and should be accompanied by quantitative assessment of CD3+ and CD20+ cell infiltration into these organs in all iMCD and control LN-injected mice.

Response

We appreciate your constructive suggestions. We added the new table and figure describing the chimerism of human hematopoietic cells in the iMCD-NOS NSG and control NSG mice to show the quantitative information (new Supplementary Table 2 and new Fig.1d). Furthermore, we presented the images of HE staining of the organs from other iMCD-NOS NSG and control mice (new Fig.1e, Supplementary Fig.2b, and Supplementary Fig.4b).

Fig. 1c, would have liked to see some quantification of weight loss over the course of experimentation, rather than representative images here. Also, it is not mentioned if there was any other clinical evidence of systemic inflammation besides weight loss/vascularity in the ear, were inflammatory markers measured?

Response

We added the new data describing the body weight change in new Fig.2c to show the detailed data of the recipient mice. We tried to check blood of the recipient mice to show the systemic inflammation *in vivo*; however, as mentioned above, the blood sampling was difficult due to the cachectic status of iMCD-NOS NSG mice and limited blood volume for detailed analysis. We appreciate your constructive suggestions.

Line 464: did you look at LN size and histology?

Response

As we responded to your question (line136-140), it is hard to detect lymphadenopathy in NSG mice. Instead of lymphadenopathy, we could observe the enlarged spleen as compared to the control mice transplanted with non-iMCD-NOS LN cells (new Supplementary Fig.3b and 3c) (line 178-180). We highly appreciate your helpful comments.

Line 473:

Figure 2 reports splenic plasmablast expansion in all iMCD LN recipients, but these data are not provided in a summary figure with statistical tests to demonstrate this expansion. Only a "representative" flow from a single mouse is shown. It is unclear whether this is seen across mice with the same LN donor, different iMCD LN donors, or control LN donors. The timepoint post-injection is also not given. Can the authors add these data to the paper?

Response

Thank you for your constructive comments. The data of iMCD-NOS NSG1, iMCD-NOS NSG2, and iMCD-NOS NSG3 were statistically processed to show the percentage of Memory B cells and plasmablast cells in new Fig.2b (line 165-168). Detailed values are described in new Supplementary Table 2. Timepoint of analysis was added in Fig.2a.

Though not shown in the flow plots, the text (page 5, line 156) reports the plasmablasts were CD20+ though plasmablasts are typically negative for CD20 (Ellebedy et al. Nat

Immunol. 2016.; Fink et al. Front. Immunol. 2012.; Covens et al. Blood. 2013.; Owczarczyk et al. Sci. Trans. Med. 2011.) Can the authors comment on this in the discussion?

Response

Thank you for your important comment. We re-analyzed the phenotypes of the engrafted human B cells based on the papers you suggested. We found that memory B cells (CD10⁻CD19⁺CD20⁺CD27⁺CD38^{dim}) and plasmablasts (CD10⁻CD19⁺CD20⁻CD27⁺CD38^{high}) expanded as compared to other B cells (non-memory B cells and non-plasmablasts) (new Fig.2b) in iMCD-NOS NSG mice. As the reviewer pointed out, plasma blasts are CD20 negative and are measured separately from memory B cells in the revised manuscript. We cited the papers and added new Fig.2b (line 162-168). Detailed values are given in new Supplementary Table 2. We highly appreciate your critical comments.

A new donor - iMCD-NOS NSG4 - is introduced here for unclear reasons. Can the authors explain this? It may be because the investigators ran out of LN cells from the other donors, but it is not clear from the text why this was done.

Response

Since we never expected that LN cells from iMCD-NOS patients engrafted and induce the systemic inflammation in our NSG mice model when we initiated the study, the amount of remaining LN cells from the initial studies (iMCD-NOS1-3) were limited. After the confirmation of reproducibility of xenotransplantation experiments from iMCD-NOS1-3 samples, we then moved to the next experiments to evaluate the differentiation of transplanted cells using LN cells from iMCD-NOS4.

The authors should have considered performing blood sampling for quantitative immunoglobulins for a broader range of donors, not just one at this unclear timepoint. IgE was assessed for unclear reasons and IgM was not measured.

Response

We appreciate your helpful suggestion. As we described above, we never expected the outcome of iMCD-NOS NSG mice when we initiated the study. We, therefore, could not evaluate the immunoglobulins of the NSG mice shown in Fig.1. We extended the analysis of immunoglobulins of the recipient mice (iMCD2,4, and 5 as shown in Fig.2 and Fig.3). Since the volume of mouse serum that we could obtain was limited (20-40 ul), we

had to select the immunoglobulins we evaluated. We measured IgE, because it has been shown to be associated with the response to IL-6 inhibition (PMID 34438448). We cited the paper and described the reason for the evaluation of IgE in the revised manuscript (line 176-177). We also added the time point of the immunoglobulins analysis in Fig.2 and Fig.3.

Line 473:

New patient (iMCD-NOS NSG4) introduced in this figure. Not mentioned in Fig.1 (although present in Supp. Table 1). Can the authors provide more information about this patient (and all patients) who provided lymph nodes for this study?

Response

According to the reviewer's helpful suggestion, we extended the information of Supplementary Table 1 including the newly added clinical samples to precisely describe the clinical information of the patients. We also added the available data of LN histology (HE staining) in new Supplementary Fig.1.

What is the timepoint for FACS images post injection? Are these expansions of T and B cells consistent across mice/patients? This was not clear from the text/figure. Can groupwise comparisons be done?

Response

We added the time point of FACS analysis in Fig.2. We also added the new table and figure describing the chimerism of human hematopoietic cells in the iMCD-NOS NSG and control NSG mice to show the detailed information (new Supplementary Table 2 and new Fig.1d). We also added new comparison analysis and detailed data of FACS-experiments (new Fig.1d, Fig.2b, Fig.2c, Fig.4d, Supplementary Fig.5a, and Supplementary Fig. 5c). We appreciate your helpful suggestions.

Line 477:

States "A representative FACS plot of cells in the spleen" - can you show compared to what is seen in the control mice instead of a representative plot?

Response

According to the reviewer's helpful suggestion. We added the new Fig.2b to show the differentiation status of engrafted human B cells (line 163-168).

Line 495:

Figure 3 reports that T-B interaction was required for the inflammatory phenotype by performing CD3-depletion on two iMCD LN donors and assessing splenic %CD45 cells and sera immunoglobulin levels. Panel A reports that the mice injected with the unmanipulated LN cells were followed for 8 weeks, but this donor seemed to induce much faster lethality in Figure 1b. Panel A is a cartoon that reports "cachexia" was observed in the unmanipulated iMCD LN recipients, but no data (e.g. weight loss, survival) is given to support the schematic. Mice who received P5 +/- CD3 depletion did not appear to have a significant difference in splenic %CD45 as the authors state in the text (page 6, line 174-5). It is also unclear how the close proximity of T/B cells in the spleen and liver shown on the immunofluorescent microscopy proves any T/B interaction as described in the text. B cells and T cells often border one another in normal lymph nodes. CD8 staining in blue is difficult to see.

Response

As you mentioned, iMCD-NOS NSG mice transplanted with unmanipulated LN cells from iMCD-NOS P2 (Fig.3) showed longer survival than NSG mice transplanted LN cells from the identical patient (Fig.1). This is presumably due to the smaller number of LN cells transplanted in the experiments shown in Fig.3 (1 million cells/mice in Fig.3 vs. 3 million cells/mice in Fig.1). These data are summarized in new Supplementary Table 2. Regarding BW changes of the recipient mice shown in Fig.3 are summarized in new Supplementary Fig.4a (line 193-194).

In addition, there was an error regarding the number of samples (P5), which has been corrected in new Fig.3b, and we found that iMCD-NOS NSG5 mice transplanted with unmanipulated LN cells exhibited significantly higher hCD45⁺ cells than NSG mice transplanted with T cell-depleted LN cells (new Fig.3b) (please refer to the response to question 4 from Reviewer#1). The color setting of the mass spectrometer was adjusted and changed to magenta, yellow, and cyan to improve the visibility. Regarding the T-B interaction, our data showed the essential role of T-B interaction and the co-localization of human T and B cells in mice for the development of iMCD-NOS like inflammation. Therefore, the detailed experiments would be necessary to clarify the detailed mechanisms underlying T-B interaction in vivo as you suggested. We would try these experiments in the future studies. We described this point in the discussion (line 317-320).

Line 501:

Figure 3B legend. Does this mean that none of the immune cells grafted?

Response

Yes. Detailed data of reconstitution of human CD45⁺ cells is now shown in the new Supplementary Table 2.

Line 512:

Figure 4 reports that Tph population is specifically expanded in iMCD and are the cellular source of CXCL13. Altogether, a larger flow panel that includes TIGIT, ICOS, CCR, and CXCL13 (with ex vivo CD3/28 re-stimulation) should be considered to characterize the Tph cells across all iMCD and control samples as had been done to better characterize these cells in other diseases where they have been implicated (Rao et al. Nature. 2017.; Uzzan et al. Nature Medicine. 2022).

Response

We presented the data of hCCR2 expression in hCD45⁺hCD3⁺hCD4⁺hPD-1^{high}hCXCR5⁻ Tph cells using multi-parameter flow immunophenotyping (new Supplementary Fig.5a) (line 217-218). Control LN cells failed in the reconstitution of human hematopoietic cells *in vivo* (new Fig.1d). Furthermore, the development of human Tph cells were not observed in the NSG mice reconstituted with normal human hematopoiesis via transplantation of normal CD34⁺ HSPCs (Fig.4b). We, therefore, could not compare the differences of Tph cells from iMCD-NOS LN cells and control subjects. We highly appreciate your constructive suggestions to improve our study.

Panel A should include percentages in gate. Panel b description (line 517 - 520) does not include which statistical tests (ANOVA?) were done: Only a simple p value is given. The color key in panel 2 is illegible.

Response

We added the percentages of the gate in Fig.4a. We apologize for the confusing figure (Fig.4b): it is a two-group comparison between the iMCD group and the control group, evaluated by the student-t test. The method of statistical analysis was added to the Figure legends of Fig.4b (line 690-692). We also modified the color key in the revised manuscript to improve the readability.

The use of only representative flow plots for CCR2 (panel c) and CXCL13 (panel g) expression is particularly unusual and should be shown for all iMCD and controls samples as groupwise comparisons with statistical tests done. Tph are certainly an excellent candidate as the cellular source of CXCL13, which has been shown to be dramatically increased in plasma of iMCD patients, but a very slight shift in the peak of one representative flow plot without showing whether it is true across samples or whether it is increased compared to control is insufficient to reach this conclusion. Furthermore, other candidates for the cellular source of iMCD (such as follicular dendritic cells) are not addressed.

Response

According to the reviewer's helpful comment, we summarized the hCCR2 expression of Tph cells in iMCD-NOS NSG mice (new Supplementary Fig. 5a) (line 217-218). Since control LN cells did not reconstitute human blood cells *in vivo* (new Supplementary Table 2 and Fig.1d) and normal CD34⁺ HSPCs did not exhibit the expansion of Tph cells *in vivo* (Fig.4b), we could not perform statistical analysis of hCCR2 expression. We evaluated the frequencies of human FDC and Tfh cells, both of which can secrete CXCL13; however, we could not detect the expansion of human FDC and Tfh cells in the recipient mice (new Supplementary Fig.5c) (line 250-253). We, therefore, focused on Tph cells as the cellular source of CXCL13 at least in our xenograft model. Of course, these data do not exclude the possibility that FDCs and Tfh cells as the cellular source of CXCL13 in the patients with iMCD-NOS. We, therefore, described the limitation of our study regarding the source of CXCL13 in iMCD-NOS (line 375-379). We also modified Fig.6 and its title to accurately show the findings observed in our PDX model (line 763-766). We highly appreciate your critical comments.

I was not able to follow why the investigators switch between spleen in the recipient mice and LN/PB characterization in the human donor even though they have only previously characterized the recipients spleens to this point. I would recommend one set of data dedicated to high-parameter flow immunophenotyping of the donor LN cells from iMCD and control LNs and another focused on flow immunophenotyping of recipient NSG mouse LN and spleen. Altogether, an orthogonal approach, such as scRNAseq and/or IF would also make sense to prove expansion of these cells in iMCD.

Response

According to the reviewer's comment, we evaluated the frequencies of CD3⁺CD4⁺PD-

1^{high}CXCR5⁻ Tph cells and CD3⁺CD4⁺PD-1^{high}CXCR5⁺ Tfh cells in the control LN cells (reactive LN cells) and iMCD-NOS LN cells using multi-parameter flow immunophenotyping and found the increased frequencies of Tph cells in iMCD-NOS LN cells (new Fig.4d) (line 219-222). We also added the data of human B cell differentiation toward memory B cells (hCD45⁺hCD10⁺hCD19⁺hCD20⁺hCD27⁺hCD38^{dim}) and plasmablasts (hCD45⁺hCD10⁺hCD19⁺hCD20⁺hCD27⁺hCD38^{high}) in iMDC-NOS NSG mice as you suggested (please refer to our response to your question line 473) using multi-parameter flow immunophenotyping (new Fig.2b) (line 162-168). The expression of hCCR2 in hCD45⁺hCD3⁺hCD4⁺hPD-1^{high}hCXCR5⁻ Tph cells were also evaluated *in vivo* (new Supplementary Fig.5a) (line 217-218). We considered that these detailed immunophenotyping you kindly recommended improved the readability of the revised manuscript. We highly appreciate your constructive suggestions.

The control that was used in panel b is confusing to me: Why are cord blood CD34⁺ HSPCs used as controls? It is not clear why control LN injected recipients (the controls used elsewhere in the paper) were not used here. SLE/RA controls would also be nice if the investigators are trying to prove specificity as they state in the figure legend title (line 514). The use of cord-blood controls instead of recipient mice injected with control LN cells is also unusual for the cytokine comparison.

Response

As we responded to your first question, LN cells from control subjects failed in the efficient reconstitution of human hematopoietic cells *in vivo*. We also extended the analysis of xenotransplantation of two reactive LN samples in the revised manuscript (new control 6-7 in Fig.1b and new Supplementary Table 2), but these reactive LN cells again failed in the efficient engraftment *in vivo* (new Supplementary Table 2 and Fig.1d). We, therefore, compared the frequencies of human Tph cells between iMCD-NOS NSG mice and NSG mice reconstituted with normal human hematopoiesis using CD34⁺ HSPCs. In general, NSG mice reconstituted with human normal hematopoiesis by transplantation of CB CD34⁺ HSPCs are commonly used as control subjects in the xenotransplantation experiments. We modified the results section to describe the reason why we employed the NSG mice transplanted with CD34⁺HSPCs as control subjects in Fig.4 (line 210-214). We again appreciate your helpful suggestions.

Surprisingly, despite being injected with LN cells from a single donor, the mice seem to

have widely different relative expression of cytokines (even CXCL13 appears down in one mouse) and no single chemokine appears to be increased in every mouse even though they are presumably all technical replicates from a single donor. Can the authors comment on this?

Response

Thank you for your constructive comments. As you pointed out, the levels of chemokines and cytokines differ in the individual recipient. These kinds of variation seem to be derived from individual differences in mice. We have observed these variations in PDX models using NSG mice as we previously reported (PMID 15920010, 21112565, 21840488, and 26551150). We described this point as one of the limitations of PDX models in the discussion (line 363-368).

I am also unclear on how comparing change in chemokines/cytokines from patient sera and the recipient mice as done in panel f helps to implicate molecules involved in disease pathogenesis: Shouldn't CXCL13 be high in the patient too since this was previously shown (Pierson. Am J Hematol. 2018.)? Where is patient 2 in therapy course (what treatments were given) / status of disease activity when sera was collected? At diagnosis/during flare? Already treated and responding? In text (page 7, line 220- 222), the authors argue that TPH cells had been shown to produce all of these cytokines in other contexts - they should consider doing an ex vivo restimulation to show whether it occurs in their model.

Response

According to the reviewer's suggestions, we modified Supplementary Table 1 and added the information of the specific treatment and CHAP score (PMID;28880697) to show the more detailed clinical information of the patients (please refer to your question line 473). All of the LN samples except for iMCD-NOS12 were obtained before the initiation of treatment (Supplementary Table 1). So, treatment was not given in the patient 2. We further confirmed the elevated levels of IFN- γ , GM-CSF, CXCL13, and TNF- α in the serum obtained at the time of biopsy of LN (P2) as compared to those in the sera of healthy individuals (n=3) (new Supplementary Fig. 5b) (line 245-247). As you suggested the ex vivo restimulation experiments would strengthen our conclusion; however, we ran out of the primary P2 LN samples. We, therefore, could not perform restimulation experiments in vitro. We appreciate your helpful suggestions.

Line 566:

Figure 5 makes the case that CXCL13 blockade can improve survival in their mouse model of iMCD. This figure is extraordinarily promising and encouraging to the field. However, I would recommend that IL6 blockade be included as this is accepted as first-line therapy for iMCD (van Rhee et al. Blood. 2018). It would also therefore be important to know whether the patients were treated with IL6 blockade and whether they responded to see how well the mouse model recapitulates the patient's disease.

Response

According to the reviewer's suggestion, we extended the xenotransplantation experiments using the LN samples obtained from the iMCD-NOS patient who exhibited exacerbation during tocilizumab treatment (new iMCD-NOS12 in Supplementary Table 1). We treated the mice using isotype IgG (control), Tocilizumab, or anti-hCXCL13 antibody. We found that there was no significant survival difference between control group and tocilizumab group, while there was a survival benefit with anti-hCXCL13 antibody treatment (new Fig. 5d and 5e) (line 281-288). These results suggest that targeting CXCL13 might be a potential therapeutic approach against iMCD patients who do not respond to IL-6 inhibition. We highly appreciate your helpful comments to improve the significance of our study.

Supplemental Table:

Clinical data provided is insufficient to understand disease severity for each patient and indeed is insufficient to establish the diagnosis as many patients included seem to not satisfy minor laboratory criteria to diagnose the disease (Fajgenbaum et al. Blood. 2017.) How the patients were treated and whether they responded would also be useful. I would also strongly recommend inclusion of representative H&E images of patient lymph nodes to prove correctness of diagnosis as there is a high degree of pathologic discordance.

Response

We extended the supplementary Table1 to describe the detailed information about iMCD-NOS patients including laboratory data and treatment. Supplementary Fig.1 shows the available histopathology images (HE staining) of surgically resected LN of each iMCD-NOS patient used in the study.

Reviewer #3 (Remarks to the Author):

In this study, Harada and colleagues established a mouse model for iMCD-NOS by engrafting LN cells from patient into immunodeficient mice. After the engraftment, the recipient mice exhibited iMCD-like fatal systemic inflammation, recapitulating the human disease. This humanized mouse model is featured by high level of human antibodies as expansion of hCD4⁺hPD-1^{high}hCXCR5⁺ hCCR2⁺ 187 Tph cells . Moreover, the authors could show that levels of hCXCL13 and some other cytokines including hIL-6 were significantly higher in mice engrafted with LN cells from patients than controls, and neutralizing hCXCL13 with anti-hCXCL13 antibody improved the survival of iMCD-NOS mice. Based on the above observations, the authors conclude that Tph cells that produce CXCL13 play a critical role in the pathogenesis of iMCD. Findings from this novel humanized mouse model shed new lights to the pathogenesis of iMCD-NOS. However, this study has several major limitations.

1. As the authors mentioned in the introduction section, a major question raised from the clinical side is that approximately half of iMCD patients do not respond to IL-6 inhibition, which indicating alternative biological pathways are involved in the pathogenesis of iMCD-NOS. A humanized mouse model provides an opportunity to address this issue. However, the study does not provide convincing evidence in this direction. Therefore, this study is not of great significance to the field in this regard.

Response

We highly appreciate your positive comments and important suggestions to improve the significance of our study. According to the reviewer's important suggestion, we extended the xenotransplantation experiments using the LN samples obtained from the iMCD-NOS patient who exhibited exacerbation during tocilizumab treatment (new iMCD-NOS12 in Supplementary Table1). We found that NSG mice treated with anti-hCXCL13 antibody exhibited significantly improved survival as compared to those treated with tocilizumab (new Fig.5d and 5e) (line 281-288). These results suggest that targeting CXCL13 might be a potential therapeutic approach against iMCD patients who do not respond to IL-6 inhibition.

2. Mice engrafted with LN cells from iMCD-NOD patients showed body weight loss, depilation , inflammation and death, which are also features of graft-versus-host disease

(GVHD). Since GVHD is common and major complication that occurs following engrafting human immune cells into immunodeficient mice (PMID: 35993192), the authors need to provide evidence for the lack of GVHD in this humanized mouse model.

Response

We highly appreciate your critical comments. In general, models of xenogeneic GVHD had been established via transplantation of higher number of ($1-5 \times 10^7$) human peripheral blood mononuclear cells (PBMNCs) (PMID: 35993192). In contrast, we transplanted less number ($0.6-3.0 \times 10^6$) of iMCD-NOS LN cells. Thus, the amount and origin of human cells differed in our experiments when compared to the previous studies of xenogeneic GVHD models. Further, we conducted additional xenotransplantation experiments to support our hypothesis that iMCD-NOS NSG mice recapitulated the disease status of iMCD-NOS. In the revised manuscript, we extended the xenotransplantation experiments using two independent reactive LN samples (new control LN6 and 7). We again confirmed that LN cells from reactive LN cells failed in the reconstitution of human hematopoiesis *in vivo* (new Fig.1d and new Supplementary Table 2). We also added the new data of memory to plasmablast differentiation of engrafted human B cells *in vivo* (new Fig.2b) (line 162-168), which represented one of the characteristics of iMCD-NOS and not for xenogeneic GVHD. We also added the data of therapeutic experiments using new iMCD-NOS LN samples (P12, please refer to your question 1), and found that about 1.1×10^6 iMCD-NOS LN cells again induced lethal inflammation *in vivo*. These data suggested that iMCD-NOS LN cells, but not reactive LN cells had a potential to recapitulate the pathogenesis of iMCD-NOS *in vivo*.

The extended xenotransplantation results kindly suggested by the reviewers including you could support our hypothesis; however, we could not completely exclude the effect of xenogeneic reaction in our model. We, therefore, cited the nice review article (PMID: 35993192) and discussed the limitation as you kindly pointed out (line 368-375).

3. The authors claimed that transfer of LN cells from iMCD patients leads to the development of lethal systemic inflammation in mice. However, in figure 1, only representative histology of lung and livers are presented. To support their statement, evaluation the histology of other organs is required, and statistical analysis need to be performed to show the difference between the two groups of mice. Due to the deficiency of T and B cells, the spleen of immunodeficient mice is tiny and featured by the lack of germinal center, and transfer of human immune cells into those mice can restore the GC (PMID: 34248959). It is interesting to know whether engraftment of LN cells from iMCD

patients and controls restore the GC in the murine spleen.

Response

According to the reviewer's helpful suggestions, we presented the data of HE staining, and immunohistochemistry analysis of iMCD-NOS NSG mice transplanted with three independent iMCD-NOS LN cells (P1, P2, and P3) in new Fig.1e and new Supplementary Fig.2b. We also showed histology of the spleen and liver from NSG mice without transplantation as negative controls in the new Fig.1e. Results of statistical analysis of the frequencies of human cells in the recipient mice between two groups are shown in new Fig.1d (line 143-147). We also showed the HE staining data of lung and kidney of the iMCD-NOS NSG2 (iMCD-NOS NSG2-10, 2-11, 2-12 for bulk and iMCD-NOS NSG2-13 for T-cell depletion) (new Supplementary Fig.4b) (line 195-198). Regarding GC formation, this is very interesting point. We observed the cluster of hCD20⁺ B cells and surrounding human T cells in the spleen of the iMCD-NOS NSG mice as shown in Fig.3. These cluster might represent GC formation. The GC formation might be consistent with our results that human B cells preferentially differentiated toward memory B cells and plasmablasts (Fig.2a and 2b) (line 162-168), *in vivo*; however, we could not detect human FDCs *in vivo* (new Supplementary Fig.5c) (line 250-253). We, therefore, considered that conclusive statement of GC formation might be overstatement based on our current data. We cited the several papers (PMID: 34248959 and 36008863) describing the usage of PDX in the study of autoimmune diseases to emphasize the usefulness of PDX models in the variety of research field (line 99-100). We highly appreciate your helpful comments to improve our study.

4. Figure 2a shows only a representative FACS plot of a mouse. Data from more mice engrafted with iMCD and control LN cells are required for quantitative statistics. In figure 2b and 2c, only mice engrafted LN cells from one patients (P4) were used for statistical analysis, which is not convincing.

Response

We highly appreciate your helpful suggestions. According to the reviewer's constructive suggestions, we added the data of B cell differentiation status in MCD-NOS NSG1-3 mice and control NSG mice (new Fig.2b) (line 162-168). We also provided the detailed data of xenotransplantation experiments in new Fig. 1d and Supplementary Table 2 (line 143-147). We added the data of body weight changes of the recipient mice (iMCD-NOS NSG2-4 and control NSG1-3) in new Fig.2c (line 174-176). Since we never expected that LN cells from iMCD-NOS patients engrafted and induce the systemic inflammation

in our NSG mice model when we initiated the study, the amount of remaining LN cells from the initial studies (iMCD-NOS 1-3) were limited. After the confirmation of reproducibility of xenotransplantation experiments from iMCD-NOS1-3 samples, we then moved to the next experiments to evaluate the differentiation of transplanted cells using LN cells from iMCD-NOS4 in Fig.2.

5. This study claims that Tph cells that produce CXCL13 play a critical role in the pathogenesis of iMCD. However, the author did not provide evidence that the Tph cells are the major source of hCXCL13.

Response

We appreciate your important comments. Regarding the source of human CXCL13, we added the several data in the revised manuscript. First, we showed that human follicular dendritic cells (FDCs) and Tfh cells which could secrete CXCL13 were almost undetectable in the iMCD-NOS NSG mice (new Supplementary Fig. 5c). We, therefore, focused on the expanded human Tph cells as the source of human CXCL13 at least in our xenograft model (line 250-255). Second, we confirmed the increased frequencies of human Tph cells in the primary LN samples obtained from the patients with iMCD-NOS (new Fig. 4d) (line 219-222). Third, we confirmed hCXCL13 expression in human Tph cells in iMCD-NOS NSG12 mice (new Supplementary Fig.6b) (line 288-290). These extended data support our hypothesis that Tph cells can be a source of CXCL13 production in iMCD-NOS NSG mice. Of course, these data do not exclude the possibility that FDCs and Tfh cells as the cellular source of CXCL13 in the patients with iMCD-NOS. We, therefore, described the limitation of our study regarding the source of CXCL13 in iMCD-NOS (line 375-379). We also modified Fig.6 and its title to accurately show the findings observed in our PDX model (line 763-766).

6. In figure 4, the difference in hCD4+hPD-1^{high}hCXCR5⁻ hCCR2⁺ 187 Tph cells and cytokines between the disease group and controls are from the comparison between mice engrafted with LN cells from iMCD patients and mice engrafted with CB-derived CD34⁺ HSPCs. From the reviewer's point of view, the control group should be mice engrafted with LN cells from patients with other disease or healthy subjects.

Response

We appreciate your constructive comments. As we mentioned above, LN cells from control subjects failed in the efficient reconstitution of human hematopoietic cells *in vivo*.

We extended xenotransplantation experiments using new reactive LN samples (new control 6 and 7) in the revised manuscript, and again confirmed that reactive LN cells (control 1,6, and 7) as well as LN cells from lymphoma samples (control 2,3,4, and 5) failed in the efficient engraftment of human cells in the recipient mice (summarized in new Supplementary Table 2 and new Fig.1d) (line 143-147). We, therefore, compared the frequencies of human Tph cells and the levels of human cytokines/chemokines between iMCD-NOS NSG mice and NSG mice reconstituted with normal human hematopoiesis using CD34⁺ HSPCs. In general, NSG mice reconstituted with human normal hematopoiesis by transplantation of normal CD34⁺ HSPCs are used as control subjects *in vivo* experiments as we previously reported (PMID 21112565, 21840488, and 26279267). We described the reason why we employed the NSG mice transplanted with CD34⁺HSPCs as control subjects in Fig.4 (line 210-214).

7. In figure 5, only survival data are presented. It is interesting to know the effect of blocking CXCL13 on other immunological, histological phenotypes.

Response

According to the reviewer's helpful suggestion, we presented the data of serum albumin level of iMCD-NOS NSG11 mice as a marker of systemic inflammation at the time of eight weeks after transplantation. The number of the available serum samples were limited (n=2). We, therefore, could not perform statistical analysis; however, the serum albumin elevated in the treated group, especially in the CXCL13 treatment group (new Supplementary Fig.6a) (line 278-281). We highly appreciate your helpful suggestion.

REVIEWER COMMENTS

Reviewer #1 (Remarks to the Author):

The authors have addressed my comments and acknowledged the limitations I have identified. I have no further comments

Reviewer #2 (Remarks to the Author):

General comments:

The authors have done a good job addressing the reviewer comments including adding more depth to characterization of samples/mice and were able to add to methods/results sections regarding control samples used (and why) and statistics used. I agree that the manuscript has been improved.

My greatest concern is about whether this xenotransplant mouse is a model of iMCD or a model of GVHD (or of both) and thus whether the findings suggests that CXCL13 inhibition is a potential treatment for iMCD or for GVHD. The lack of engraftment of control LNs (and absence of human immune cells present in these mice) means that the changes observed could either be specific to iMCD or non-specific changes associated with xenorejection or alloimmunity. Though there could be some specific factors/features of iMCD that enables these non-malignant lymphocytes to engraft, that still does no help with teasing apart whether this is a model of iMCD or xenorejection/alloimmunity.

To help address whether this is a model of iMCD, in their revised manuscript, the authors provide additional histology and, for a single unspecified time point, weights. It remains unclear to me that they are describing Castleman in these mice and I do not believe the provided histology, weights, or photographs of the spleen help to distinguish from xenogeneic GVHD, which can occur at the cell doses used in this experiment. The fact that only iMCD donor lymph nodes and not lymph nodes from control conditions were capable of engrafting is curious and a confounder for all other correlative data presented here.

The authors' primary argument against this representing GVHD being that they used fewer cells was not a convincing argument unless they have data on a similar number of cells not causing GVHD in a murine model. The lack of engraftment of control samples really means that the study is without a control to determine if the changes are iMCD specific or engraftment specific.

One way to address this would be to evaluate histology and other features typically found in humans with iMCD to determine if they're shared in these mice. Another way to potentially address this would be to perform an experiment where lymph node is infused from another disease state and the cells engraft, treat them with anti-CXCL13, and then compare those mice to these "iMCD mice." This would help to tease apart whether the features are due to engraftment of human cells or a model of iMCD. Another way to potentially address this would be to include in the title/abstract/findings that this may be due to xenorejection or alloimmunity and thus possibly a model of GVHD.

My concerns related to the controls:

Figures 1-3 - could be GVHD effect.

Figure 4 - HSPCs vs. iMCD LN cells is difficult comparison to make because they are so different.

Figure 4b - Its interesting that these cells are so highly represented in the spleen among the CD4+ fraction, but I don't think its fair to say they expanded in comparison to HSPCs that didn't.

I also have some concerns related to the iMCD cases. In their revised manuscript, the authors provided clinical images and a supplementary table with clinical data. This is a great improvement, though it remains difficult to confirm the iMCD diagnosis for a number of these cases where the H&E image is not clearly diagnostic (e.g. iMCD-NOS2) or the clinical parameters do not show evidence of minor iMCD criteria (e.g. iMCD-3).

Additional comments:

1. Fig 1e: why are the images provided at different magnifications between neg control and cases? Can you provide the same mag and include that in the legend?

2. Can the authors hypothesize why LN cells from lymphoma samples are failing to engraft? Or if there is another lymphoma model that this could be done with? What did you previous use to generate PDX models?

3. I appreciate the authors sharing that the CD34+ HSPCs were uses as controls because the non-iMCD LNs failed to engraft, but did the CD34+ HSPCs engraft? If so, can you provide this data potentially like in Fig 4d?

4. Fig 4d-g – Very interesting to see increased Tph cells in LN cells of patients and CXCL13 production (Fig 4g). Is there a reason by intracellular flow was just provided for 1 patient in Fig 4g and not other patients as well?

5. I understand that B cell depletion in an new experiment is infeasible, but I think you should mention in the discussion section (2nd paragraph) that B cell depletion studies are needed.

6. Fig 2b – shouldn't the comparison be between the two states (control vs iMCD) rather than between B cell types within the cases?

7. What do the authors mean in their responses that "we never expected that LN cells from iMCD-NOS patients engrafted and induce the systemic inflammation in our NSG mice model when we initiated the study" – wasn't this the hypothesis? Or did you perform the engraftment for another reason?

Reviewer #3 (Remarks to the Author):

All my questions and concerns have been well addressed or discussed.

We highly appreciate the editor and the reviewers for positive comments and their valuable feedback. We thank reviewer#2 for the suggestions to further improve our study. We have responded to the reviewer(R2)'s concerns in a point-by-point manner and have accordingly revised our manuscript. We believe the addition of the new data suggested by the reviewer#2 improves substantially the impact of our work.

REVIEWER COMMENTS

Reviewer #1 (Remarks to the Author):

The authors have addressed my comments and acknowledged the limitations I have identified. I have no further comments

Reviewer #2 (Remarks to the Author):

General comments:

The authors have done a good job addressing the reviewer comments including adding more depth to characterization of samples/mice and were able to add to methods/results sections regarding control samples used (and why) and statistics used. I agree that the manuscript has been improved.

Response: First, we highly appreciate your positive and constructive comments in the first revision. We further tried to improve our current study according to your additional helpful comments. Please see the point-by-point response below.

My greatest concern is about whether this xenotransplant mouse is a model of iMCD or a model of GVHD (or of both) and thus whether the findings suggests that CXCL13 inhibition is a potential treatment for iMCD or for GVHD. The lack of engraftment of control LNs (and absence of human immune cells present in these mice) means that the changes observed could either be specific to iMCD or non-specific changes associated with xenorejection or alloimmunity. Though there could be some specific factors/features of iMCD that enables these non-malignant lymphocytes to engraft, that still does no help with teasing apart whether this is a model of iMCD or xenorejection/alloimmunity.

Response: Immunodeficient mice have been developed and modified to prevent xenorejection by depleting lymphoid lineage cells and inducing macrophage tolerance. NSG mice are one of the most utilized types of immunodeficient mice across diverse research fields. In the case of xenorejection, all human samples, including iMCD-NOS LN cells and CD34⁺ HSPCs, would be equally rejected. Additionally, we noted adequate engraftment of human CD45⁺ cells (>5% *in vivo*) in three out of 42 control NSG mice (Supplemental Table 2) (please see our comment below), implying that

xenorejection may not explain our results, in line with numerous previous studies employing NSG mice. Regarding the alloimmunity you mentioned, we transplanted LN cells from a single patient into all NSG mice; the reconstituted human cells were derived from a single donor. Consequently, an allogeneic reaction could not occur due to the absence of allogeneic cells (cells from a distinct human donor) in the NSG mice. We appreciate your comments.

To help address whether this is a model of iMCD, in their revised manuscript, the authors provide additional histology and, for a single unspecified time point, weights. It remains unclear to me that they are describing Castleman in these mice and I do not believe the provided histology, weights, or photographs of the spleen help to distinguish from xenogeneic GVHD, which can occur at the cell doses used in this experiment. The fact that only iMCD donor lymph nodes and not lymph nodes from control conditions were capable of engrafting is curious and a confounder for all other correlative data presented here.

Response: In general, reconstituting human mature lymphoid malignancies in immunodeficient mice through the transplantation of primary cells from the patients is challenging without specific modifications such as local injection into implanted human bones, flanks, omentum, or under the kidney capsule (PMID 28348046, 29296768, 26773040, 35992875). Even with these modifications, success rates remain modest. Therefore, we believe that the efficient engraftment and expansion of iMCD-NOS LN cells without specific modifications signifies unique characteristics of iMCD-NOS LN cells as you mentioned. We added this point in the discussion of the revised manuscript (line 357-361).

Regarding xenogeneic GVHD, lymphoma PDX models reconstituted with primary LN samples using special modifications described above did not develop systemic inflammation resembling xenogeneic GVHD, and they survived long enough for the evaluation of tumor propagation, despite the engraftment and expansion of hCD45⁺ cells. Consistent with these observations, three NSG mice out of 42 control mice that exhibited sufficient engraftment did not develop systemic inflammation and survived long-term (Supplemental Table 2) (line 147-149). Furthermore, humanized NSG mice reconstituted with hCD34⁺ HSPCs did not trigger xenogeneic GVHD *in vivo*, even though these mice exhibited equivalent frequencies of hCD45⁺ cells as compared to iMCD-NOS NSG2 (new Supplemental Fig. 5c). Therefore, the engraftment and expansion of human blood cells in the immunodeficient mice do not always correlate with the development of xenogeneic GVHD *in vivo*, and the induction of systemic inflammation appears restricted to iMCD-NOS NSG mice. Hence, we consider that the development of lethal inflammation seems to be specific to iMCD-NOS NSG mice (line 380-387). Furthermore, our PDX model showed the expansion of human B cells and preferential differentiation toward memory/plasmablasts within the B cell lineage (Fig. 2 and 3). The impairment

of B cell development and differentiation represents important characteristics of GVHD in both human and mouse (PMID 11435323, 20354171, 24833353, and 30228128). We, therefore, considered that our PDX model recapitulated iMCD-like systemic inflammation rather than GVHD at least in terms of the B cell expansion and differentiation (line 388-392). We highly appreciate your helpful comments.

The authors' primary argument against this representing GVHD being that they used fewer cells was not a convincing argument unless they have data on a similar number of cells not causing GVHD in a murine model. The lack of engraftment of control samples really means that the study is without a control to determine if the changes are iMCD specific or engraftment specific.

One way to address this would be to evaluate histology and other features typically found in humans with iMCD to determine if they're shared in these mice. Another way to potentially address this would be to perform an experiment where lymph node is infused from another disease state and the cells engraft, treat them with anti-CXCL13, and then compare those mice to these "iMCD mice." This would help to tease apart whether the features are due to engraftment of human cells or a model of iMCD. Another way to potentially address this would be to include in the title/abstract/findings that this may be due to xenorejection or alloimmunity and thus possibly a model of GVHD.

Response: We highly appreciate your constructive comments. As we responded to questions above, there is no factor regarding xenorejection and alloimmunity in our results. It is, however, impossible to quantitatively measure the extent to which xenogeneic GVHD contributes to the systemic inflammation and the effect of xenogeneic GVHD cannot be completely ruled out. We, therefore, modified the discussion describing xenogeneic GVHD according to your helpful suggestions (line 373-392).

My concerns related to the controls:

Figures 1-3 - could be GVHD effect.

Figure 4 – HSPCs vs. iMCD LN cells is difficult comparison to make because they are so different.

Figure 4b – Its interesting that these cells are so highly represented in the spleen among the CD4+ fraction, but I don't think its fair to say they expanded in comparison to HSPCs that didn't.

Response: Regarding Fig.1, as we responded to your questions, three out of 42 control NSG mice reconstituted with sufficient hCD45⁺ cells did not develop lethal systemic inflammation and survived (Fig.1 and Supplemental Table 2). In addition, humanized NSG mice reconstituted with hCD34⁺ HSPCs also did not trigger xenogeneic GVHD *in vivo*, even though hCD45⁺ chimerism was equivalent to iMCD-NOS NSG mice (new Supplemental Fig. 5c). These results suggested that the induction of

systemic inflammation seems to be specific to iMCD-NOS NSG mice. It is, however, impossible to quantitatively measure the extent to which xenogeneic GVHD contributes to the systemic inflammation and the effect of xenogeneic GVHD cannot be completely ruled out. We, therefore, discussed the limitation of this point according to your helpful suggestions (line 373-392).

Regarding Fig.2-3, B cell expansion and differentiation represent effects contrary to GVHD (PMID 11435323, 20354171, 24833353, and 30228128), as we mentioned above. We highlighted this point to emphasize the significance of B cell development in our iMCD-NOS NSG mice as you pointed out (line 388-392).

Regarding Fig.4, we did not directly compare CD34⁺ HSPCs and iMCD-NOS LN cells. Rather, we compared the mature human hematopoietic cells that were reconstituted with CD34⁺ HSPCs and iMCD-NOS LN cells after xenotransplantation. The frequencies of hCD45⁺, hCD3⁺, and hCD19⁺ cells did not differ between the two groups (new Supplemental Fig.5c) (line 239-241). Therefore, we consider the comparison to be theoretically appropriate. Moreover, using humanized NSG mice reconstituted with CD34⁺ HSPCs as control subjects is a standard method in the studies using PDX models (PMID 23059428, 27959627, 31726047, and 34869042) (line 235-237). Thus, it is difficult to avoid using humanized NSG mice as control subjects.

I also have some concerns related to the iMCD cases. In their revised manuscript, the authors provided clinical images and a supplementary table with clinical data. This is a great improvement, though it remains difficult to confirm the iMCD diagnosis for a number of these cases where the H&E image is not clearly diagnostic (e.g. iMCD-NOS2) or the clinical parameters do not show evidence of minor iMCD criteria (e.g. iMCD-3).

Response: We prepared new H&E image (iMCD-NOS2) and presented in new Supplementary Fig.1. Regarding the clinical data of iMCD-NOS3, we re-checked the clinical data of the corresponding patient and found that the presented data was obtained after the treatment with corticosteroid. We, therefore, presented the data of iMCD-NOS3 obtained before treatment in new Supplementary Table1. We highly appreciate your helpful suggestions.

Additional comments:

1. Fig 1e: why are the images provided at different magnifications between neg control and cases? Can you provide the same mag and include that in the legend?

Response: The images are provided at the same magnifications (scale bar is automatically added by the microscope). Since the spleen of negative control is very small as shown in Supplemental Fig. 3b, it seems different magnitude when compared to that of iMCD-NOS NSG mice. Original images are

provided at the public database as described in the section of Data availability.

2. Can the authors hypothesize why LN cells from lymphoma samples are failing to engraft? Or if there is another lymphoma model that this could be done with? What did you previous use to generate PDX models?

Response: We appreciate your comment. As we mentioned above, reconstituting human lymphoma/myeloma in immunodeficient mice through the transplantation of primary cells from patients is challenging without specific modifications such as local injection into implanted human bones, flanks, or under the kidney capsule (PMID 28348046, 29296768, 26773040, 35992875). Even with these modifications, success rates remain modest. In addition to lymphoma/myeloma, 1×10^7 chronic lymphocytic leukemia cells completely failed in the engraftment in NSG mice as we previously reported (PMID 21840488). We have generated PDX models using normal CD34⁺ HSPCs and leukemic stem cells of acute leukemia in the previous studies (PMID 21112565, 21840488 and 26279267). Thus, PDX models have been intensively utilized in the research field of normal and malignant stem cells.

3. I appreciate the authors sharing that the CD34⁺ HSPCs were uses as controls because the non-iMCD LNs failed to engraft, but did the CD34⁺ HSPCs engraft? If so, can you provide this data potentially like in Fig 4d?

Response: According to the reviewer's helpful suggestions, we added the detailed data of human hematopoietic reconstitution when normal CD34⁺ HSPCs were transplanted in new supplemental Fig.5c.

4. Fig 4d-g – Very interesting to see increased Tph cells in LN cells of patients and CXCL13 production (Fig 4g). Is there a reason by intracellular flow was just provided for 1 patient in Fig 4g and not other patients as well?

Response: According to the reviewer's helpful suggestions, we added the data of intracellular FCM analysis of CXCL13 expression in Tph cells using the iMCD-NOS LN (P1) sample. (Fig. 4g) (line 255) We appreciate your comment.

5. I understand that B cell depletion in an new experiment is infeasible, but I think you should mention in the discussion section (2nd paragraph) that B cell depletion studies are needed.

Response: We highly appreciate your helpful suggestions. According to the comment, we described the necessity of B cell depletion experiments to accurately show that B-T interaction is required for the pathogenesis of iMCD-NOS (line318-321).

6. Fig 2b – shouldn't the comparison be between the two states (control vs iMCD) rather than between B cell types within the cases?

Response: We highly appreciate your helpful suggestions. According to the comment, we added new Supplemental Fig 3a which shows the comparison of memory B cells or plasmablasts fraction between the two states (control vs. iMCD).

7. What do the authors mean in their responses that “we never expected that LN cells from iMCD-NOS patients engrafted and induce the systemic inflammation in our NSG mice model when we initiated the study” – wasn't this the hypothesis? Or did you perform the engraftment for another reason?

Response: Related to your question #2, reconstituting human mature lymphoid malignancies in immunodeficient mice via the transplantation of primary cells from the patients proves challenging without specific modifications such as local injections into implanted human bones, flanks, omentum, or underneath the kidney capsule (PMID 28348046, 29296768, 26773040, 35992875). Even with these alterations, success rates remain modest. From our extensive research on human mature lymphoid malignancies (PMID 21840488, 30194138, and 34638128), we understand the difficulties of reconstituting human lymphoid tissues *in vivo*, as evidenced by our previous studies. Consequently, when initiating the current study, we did not anticipate that the tail vein injection of iMCD-NOS LN cells would result in efficient engraftment and expansion.

Reviewer #3 (Remarks to the Author):

All my questions and concerns have been well addressed or discussed.

REVIEWERS' COMMENTS

Reviewer #2 (Remarks to the Author):

Thank you for your responses and revisions of the article. While I am still not certain that the findings in this model are entirely representative of MCD-specific features and not due to non-specific xenorejection since the other xenotransplanted tissue did not engraft in the control mice, I do think the authors have addressed my specific concerns and feel the manuscript is ready for publication.

We highly appreciate the editor and the reviewers for positive comments and their valuable feedback.

REVIEWER COMMENTS

Reviewer #2 (Remarks to the Author):

Thank you for your responses and revisions of the article. While I am still not certain that the findings in this model are entirely representative of MCD-specific features and not due to non-specific xenorejection since the other xenotransplanted tissue did not engraft in the control mice, I do think the authors have addressed my specific concerns and feel the manuscript is ready for publication.

Response

We highly appreciate your valuable feedback and helpful suggestions to improve our study.